# QPLEX: Duplex Dueling Multi-Agent Q-Learning

**Jianhao Wang**[*1]**, Zhizhou Ren**[*1]**, Terry Liu**[1]**, Yang Yu**[2]**, Chongjie Zhang**[1]
[1]Institute for Interdisciplinary Information Sciences, Tsinghua University, China
[2]Polixir Technologies, China
`{wjh19, rzz16, liudr18}@mails.tsinghua.edu.cn`
`yuy@nju.edu.cn`
`chongjie@tsinghua.edu.cn`

## Abstract

We explore value-based multi-agent reinforcement learning (MARL) in the popular paradigm of centralized training with decentralized execution (CTDE). CTDE has an important concept, *Individual-Global-Max* (IGM) principle, which requires the consistency between joint and local action selections to support efficient local decision-making. However, in order to achieve scalability, existing MARL methods either limit representation expressiveness of their value function classes or relax the IGM consistency, which may suffer from instability risk or may not perform well in complex domains. This paper presents a novel MARL approach, called *duPLEX dueling multi-agent Q-learning* (QPLEX), which takes a duplex dueling network architecture to factorize the joint value function. This duplex dueling structure encodes the IGM principle into the neural network architecture and thus enables efficient value function learning. Theoretical analysis shows that QPLEX achieves a complete IGM function class. Empirical experiments on StarCraft II micromanagement tasks demonstrate that QPLEX significantly outperforms state-of-the-art baselines in both online and offline data collection settings, and also reveal that QPLEX achieves high sample efficiency and can benefit from offline datasets without additional online exploration[1].

## 1 Introduction

Cooperative multi-agent reinforcement learning (MARL) has broad prospects for addressing many complex real-world problems, such as sensor networks (Zhang & Lesser, 2011), coordination of robot swarms (Hüttenrauch et al., 2017), and autonomous cars (Cao et al., 2012). However, cooperative MARL encounters two major challenges of scalability and partial observability in practical applications. The joint state-action space grows exponentially as the number of agents increases. The partial observability and communication constraints of the environment require each agent to make its individual decisions based on local action-observation histories. To address these challenges, a popular MARL paradigm, called *centralized training with decentralized execution* (CTDE) (Oliehoek et al., 2008; Kraemer & Banerjee, 2016), has recently attracted great attention, where agents' policies are trained with access to global information in a centralized way and executed only based on local histories in a decentralized way.

Many CTDE learning approaches have been proposed recently, among which value-based MARL algorithms (Sunehag et al., 2018; Rashid et al., 2018; Son et al., 2019; Wang et al., 2019b) have shown state-of-the-art performance on challenging tasks, e.g., unit micromanagement in StarCraft II (Samvelyan et al., 2019). To enable effective CTDE for multi-agent Q-learning, it is critical that the joint greedy action should be equivalent to the collection of individual greedy actions of agents, which is called the IGM (*Individual-Global-Max*) principle (Son et al., 2019). This IGM principle provides two advantages: 1) ensuring the policy consistency during centralized training (learning the joint Q-function) and decentralized execution (using individual Q-functions) and 2) enabling

---

[*]Equal contribution.
[1]Videos available at `https://sites.google.com/view/qplex-marl/`.

scalable centralized training of computing one-step TD target of the joint Q-function (deriving joint greedy action selection from individual Q-functions). To realize this principle, VDN (Sunehag et al., 2018) and QMIX (Rashid et al., 2018) propose two sufficient conditions of IGM to factorize the joint action-value function. However, these two decomposition methods suffer from structural constraints and limit the joint action-value function class they can represent. As shown by Wang et al. (2020a), the incompleteness of the joint value function class may lead to poor performance or potential risk of training instability in the offline setting (Levine et al., 2020). Several methods have been proposed to address this structural limitation. QTRAN (Son et al., 2019) constructs two soft regularizations to align the greedy action selections between the joint and individual value functions. WQMIX (Rashid et al., 2020) considers a weighted projection that places more importance on better joint actions. However, due to computational considerations, both their implementations are approximate and based on heuristics, which cannot guarantee the IGM consistency exactly. Therefore, achieving the complete expressiveness of the IGM function class with effective scalability remains an open problem for cooperative MARL.

To address this challenge, this paper presents a novel MARL approach, called *duPLEX dueling multi-agent Q-learning* (QPLEX), that takes a duplex dueling network architecture to factorize the joint action-value function into individual action-value functions. QPLEX introduces the dueling structure $Q = V + A$ (Wang et al., 2016) for representing both joint and individual (duplex) action-value functions and then reformalizes the IGM principle as an *advantage-based IGM*. This reformulation transforms the IGM consistency into the constraints on the value range of the advantage functions and thus facilitates the action-value function learning with linear decomposition structure. Different from QTRAN and WQMIX (Son et al., 2019; Rashid et al., 2020) losing the guarantee of exact IGM consistency due to approximation, QPLEX takes advantage of a duplex dueling architecture to encode it into the neural network structure and provide a guaranteed IGM consistency. To our best knowledge, QPLEX is the first multi-agent Q-learning algorithm that effectively achieves high scalability with a full realization of the IGM principle.

We evaluate the performance of QPLEX in both didactic problems proposed by prior work (Son et al., 2019; Wang et al., 2020a) and a range of unit micromanagement benchmark tasks in StarCraft II (Samvelyan et al., 2019). In these didactic problems, QPLEX demonstrates its full representation expressiveness, thereby learning the optimal policy and avoiding the potential risk of training instability. Empirical results on more challenging StarCraft II tasks show that QPLEX significantly outperforms other multi-agent Q-learning baselines in online and offline data collections. It is particularly interesting that QPLEX shows the ability to support offline training, which is not possessed by other baselines. This ability not only provides QPLEX with high stability and sample efficiency but also with opportunities to efficiently utilize multi-source offline data without additional online exploration (Fujimoto et al., 2019; Fu et al., 2020; Levine et al., 2020; Yu et al., 2020).

## 2 PRELIMINARIES

### 2.1 DECENTRALIZED PARTIALLY OBSERVABLE MDP (DEC-POMDP)

We model a fully cooperative multi-agent task as a Dec-POMDP (Oliehoek et al., 2016) defined by a tuple $\mathcal{M} = \langle \mathcal{N}, \mathcal{S}, \mathcal{A}, P, \Omega, O, r, \gamma \rangle$, where $\mathcal{N} \equiv \{1, 2, \ldots, n\}$ is a finite set of agents and $s \in \mathcal{S}$ is a finite set of global states. At each time step, every agent $i \in \mathcal{N}$ chooses an action $a_i \in \mathcal{A} \equiv \{\mathcal{A}^{(1)}, \ldots, \mathcal{A}^{(|\mathcal{A}|)}\}$ on a global state $s$, which forms a joint action $\boldsymbol{a} \equiv [a_i]_{i=1}^n \in \boldsymbol{\mathcal{A}} \equiv \mathcal{A}^n$. It results in a joint reward $r(s, \boldsymbol{a})$ and a transition to the next global state $s' \sim P(\cdot|s, \boldsymbol{a})$. $\gamma \in [0, 1)$ is a discount factor. We consider a *partially observable* setting, where each agent $i$ receives an individual partial observation $o_i \in \Omega$ according to the observation probability function $O(o_i|s, a_i)$. Each agent $i$ has an action-observation history $\tau_i \in \mathcal{T} \equiv (\Omega \times \mathcal{A})^*$ and constructs its individual policy $\pi_i(a|\tau_i)$ to jointly maximize team performance. We use $\boldsymbol{\tau} \in \boldsymbol{\mathcal{T}} \equiv \mathcal{T}^n$ to denote joint action-observation history. The formal objective function is to find a joint policy $\boldsymbol{\pi} = \langle \pi_1, \ldots, \pi_n \rangle$ that maximizes a joint value function $V^{\boldsymbol{\pi}}(s) = \mathbb{E}\left[\sum_{t=0}^{\infty} \gamma^t r_t | s_0 = s, \boldsymbol{\pi}\right]$. Another quantity of interest in policy search is the joint action-value function $Q^{\boldsymbol{\pi}}(s, \boldsymbol{a}) = r(s, \boldsymbol{a}) + \gamma \mathbb{E}_{s'}[V^{\boldsymbol{\pi}}(s')]$.

### 2.2 DEEP MULTI-AGENT Q-LEARNING IN DEC-POMDP

Q-learning algorithms is a popular algorithm to find the optimal joint action-value function $Q^*(s, \boldsymbol{a}) = r(s, \boldsymbol{a}) + \gamma \mathbb{E}_{s'}[\max_{\boldsymbol{a}'} Q^*(s', \boldsymbol{a}')]$. Deep Q-learning represents the action-value function

with a deep neural network parameterized by $\boldsymbol{\theta}$. Mutli-agent Q-learning algorithms (Sunehag et al., 2018; Rashid et al., 2018; Son et al., 2019; Yang et al., 2020) use a replay memory $D$ to store the transition tuple $(\boldsymbol{\tau}, \boldsymbol{a}, r, \boldsymbol{\tau}')$, where $r$ is the reward for taking action $\boldsymbol{a}$ at joint action-observation history $\boldsymbol{\tau}$ with a transition to $\boldsymbol{\tau}'$. Due to partial observability, $Q(\boldsymbol{\tau}, \boldsymbol{a}; \boldsymbol{\theta})$ is used in place of $Q(s, \boldsymbol{a}; \boldsymbol{\theta})$. Thus, parameters $\boldsymbol{\theta}$ are learnt by minimizing the following expected TD error:

$$\mathcal{L}(\boldsymbol{\theta}) = \mathbb{E}_{(\boldsymbol{\tau}, \boldsymbol{a}, r, \boldsymbol{\tau}') \in D} \left[ \left( r + \gamma V\left(\boldsymbol{\tau}'; \boldsymbol{\theta}^-\right) - Q(\boldsymbol{\tau}, \boldsymbol{a}; \boldsymbol{\theta}) \right)^2 \right], \qquad (1)$$

where $V\left(\boldsymbol{\tau}'; \boldsymbol{\theta}^-\right) = \max_{\boldsymbol{a}'} Q\left(\boldsymbol{\tau}', \boldsymbol{a}'; \boldsymbol{\theta}^-\right)$ is the one-step expected future return of the TD target and $\boldsymbol{\theta}^-$ are the parameters of the target network, which will be periodically updated with $\boldsymbol{\theta}$.

## 2.3 CENTRALIZED TRAINING WITH DECENTRALIZED EXECUTION (CTDE)

CTDE is a popular paradigm of cooperative multi-agent deep reinforcement learning (Sunehag et al., 2018; Rashid et al., 2018; Wang et al., 2019a; 2020b;c;d). Agents are trained in a centralized way and granted access to other agents' information or the global states during the centralized training process. However, due to partial observability and communication constraints, each agent makes its own decision based on its local action-observation history during the decentralized execution phase. IGM (*Individual-Global-Max*; Son et al., 2019) is a popular principle to realize effective value-based CTDE, which asserts the consistency between joint and local greedy action selections in the joint action-value $Q_{tot}(\boldsymbol{\tau}, \boldsymbol{a})$ and individual action-values $[Q_i(\tau_i, a_i)]_{i=1}^n$:

$$\forall \boldsymbol{\tau} \in \boldsymbol{\mathcal{T}}, \underset{\boldsymbol{a} \in \boldsymbol{\mathcal{A}}}{\arg\max}\, Q_{tot}(\boldsymbol{\tau}, \boldsymbol{a}) = \left( \underset{a_1 \in \mathcal{A}}{\arg\max}\, Q_1(\tau_1, a_1), \ldots, \underset{a_n \in \mathcal{A}}{\arg\max}\, Q_n(\tau_n, a_n) \right). \qquad (2)$$

Two factorization structures, **additivity** and **monotonicity**, has been proposed by VDN (Sunehag et al., 2018) and QMIX (Rashid et al., 2018), respectively, as shown below:

$$Q_{tot}^{\text{VDN}}(\boldsymbol{\tau}, \boldsymbol{a}) = \sum_{i=1}^n Q_i(\tau_i, a_i) \quad \text{and} \quad \forall i \in \mathcal{N}, \frac{\partial Q_{tot}^{\text{QMIX}}(\boldsymbol{\tau}, \boldsymbol{a})}{\partial Q_i(\tau_i, a_i)} > 0.$$

Qatten (Yang et al., 2020) is a variant of VDN, which supplements global information through a multi-head attention structure. It is known that, these structures implement sufficient but not necessary conditions for the IGM constraint, which limit the representation expressiveness of joint action-value functions (Mahajan et al., 2019). There exist tasks whose factorizable joint action-value functions can not be represented by these decomposition methods, as shown in Section 4. In contrast, QTRAN (Son et al., 2019) transforms IGM into a linear constraint and uses it as soft regularization constraints. WQMIX (Rashid et al., 2020) introduces a weighting mechanism into the projection of monotonic value factorization, in order to place more importance on better joint actions. However, these relaxations may violate the exact IGM consistency and may not perform well in complex problems.

## 3 QPLEX: DUPLEX DUELING MULTI-AGENT Q-LEARNING

In this section, we will first introduce advantage-based IGM, equivalent to the regular IGM principle, and, with this new definition, convert the IGM consistency of greedy action selection to simple constraints on advantage functions. We then present a novel deep MARL model, called *duPLEX dueling multi-agent Q-learning algorithm* (QPLEX), that directly realizes these constraints by a scalable neural network architecture.

### 3.1 ADVANTAGE-BASED IGM

To ensure the consistency of greedy action selection on the joint and local action-value functions, the IGM principle constrains the relative order of Q-values over actions. From the perspective of dueling decomposition structure $Q = V + A$ proposed by Dueling DQN (Wang et al., 2016), this consistency should only constrain the action-dependent advantage term $A$ and be free of the state-value function $V$. This observation naturally motivates us to reformalize the IGM principle as advantage-based IGM, which transforms the consistency constraint onto advantage functions.

**Definition 1** (Advantage-based IGM). *For a joint action-value function $Q_{tot}: \boldsymbol{\mathcal{T}} \times \boldsymbol{\mathcal{A}} \mapsto \mathbb{R}$ and individual action-value functions $[Q_i : \mathcal{T} \times \mathcal{A} \mapsto \mathbb{R}]_{i=1}^n$, where $\forall \boldsymbol{\tau} \in \boldsymbol{\mathcal{T}}, \forall \boldsymbol{a} \in \boldsymbol{\mathcal{A}}, \forall i \in \mathcal{N}$,*

$$\textbf{\textit{(Joint Dueling)}} \quad Q_{tot}(\boldsymbol{\tau}, \boldsymbol{a}) = V_{tot}(\boldsymbol{\tau}) + A_{tot}(\boldsymbol{\tau}, \boldsymbol{a}) \text{ and } V_{tot}(\boldsymbol{\tau}) = \max_{\boldsymbol{a}'} Q_{tot}(\boldsymbol{\tau}, \boldsymbol{a}'), \quad (3)$$

$$\textbf{\textit{(Individual Dueling)}} \quad Q_i(\tau_i, a_i) = V_i(\tau_i) + A_i(\tau_i, a_i) \text{ and } V_i(\tau_i) = \max_{a_i'} Q_i(\tau_i, a_i'), \qquad (4)$$

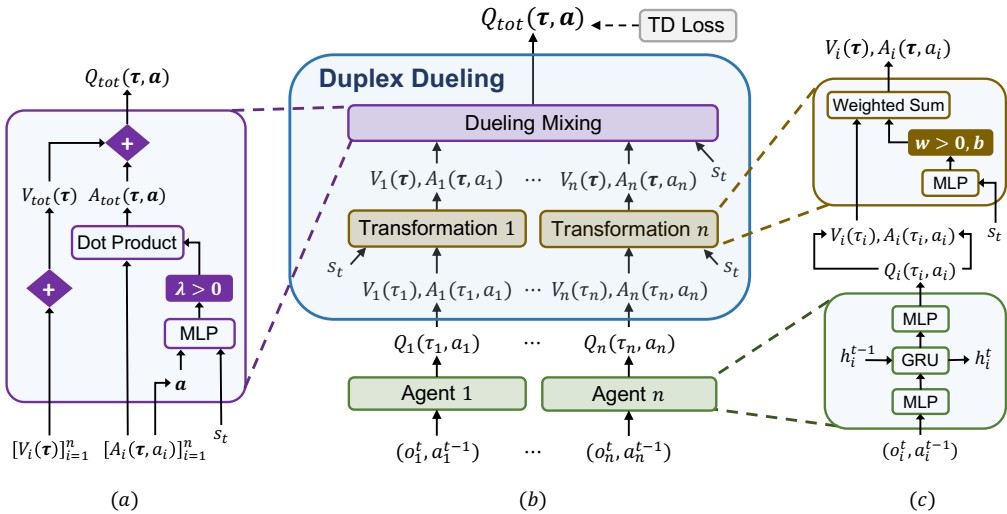

Figure 1: (a) The dueling mixing network structure. (b) The overall QPLEX architecture. (c) Agent network structure (bottom) and Transformation network structure (top).

*such that the following holds*

$$\arg\max_{\boldsymbol{a}\in\mathcal{A}} A_{tot}(\boldsymbol{\tau},\boldsymbol{a}) = \left(\arg\max_{a_1\in\mathcal{A}} A_1(\tau_1,a_1),\dots,\arg\max_{a_n\in\mathcal{A}} A_n(\tau_n,a_n)\right), \tag{5}$$

*then, we can say that $[Q_i]_{i=1}^n$ satisfies advantage-based IGM for $Q_{tot}$.*

As specified in Definition 1, advantage-based IGM takes a duplex dueling architecture, *Joint Dueling* and *Individual Dueling*, which induces the joint and local (duplex) advantage functions by $A = Q - V$. Compared with regular IGM, advantage-based IGM transfers the consistency constraint on action-value functions stated in Eq. (2) to that on advantage functions. This change is an equivalent transformation because the state-value terms $V$ do not affect the action selection, as shown by Proposition 1.

**Proposition 1.** *The advantage-based IGM and IGM function classes are equivalent.*

One key benefit of using advantage-based IGM is that its consistency constraint can be directly realized by limiting the value range of advantage functions, as indicated by the following fact.

**Fact 1.** *The constraint of advantage-based IGM stated in Eq. (5) is equivalent to that when $\forall \boldsymbol{\tau} \in \mathcal{T}$, $\forall \boldsymbol{a}^* \in \mathcal{A}^*(\boldsymbol{\tau})$, $\forall \boldsymbol{a} \in \mathcal{A} \setminus \mathcal{A}^*(\boldsymbol{\tau})$, $\forall i \in \mathcal{N}$,*

$$A_{tot}(\boldsymbol{\tau},\boldsymbol{a}^*) = A_i(\tau_i,a_i^*) = 0 \quad and \quad A_{tot}(\boldsymbol{\tau},\boldsymbol{a}) < 0, A_i(\tau_i,a_i) \leq 0, \tag{6}$$

*where $\mathcal{A}^*(\boldsymbol{\tau}) = \{\boldsymbol{a}|\boldsymbol{a} \in \mathcal{A}, Q_{tot}(\boldsymbol{\tau},\boldsymbol{a}) = V_{tot}(\boldsymbol{\tau})\}$.*

To achieve a full expressiveness power of advantage-based IGM or IGM, Fact 1 enables us to develop an efficient MARL algorithm that allows the joint state-value function learning with any scalable decomposition structure and just imposes simple constraints limiting value ranges of advantage functions. The next subsection will describe such a MARL algorithm.

### 3.2 THE QPLEX ARCHITECTURE

In this subsection, we present a novel multi-agent Q-learning algorithm with a duplex dueling architecture, called QPLEX, which exploits Fact 1 and realizes the advantage-based IGM constraint. The overall architecture of QPLEX is illustrated in Figure 1, which consists of two main components as follows: (i) an *Individual Action-Value Function* for each agent, and (ii) a *Duplex Dueling* component that composes individual action-value functions into a joint action-value function under the advantage-based IGM constraint. During the centralized training, the whole network is learned in an end-to-end fashion to minimize the TD loss as specified in Eq. (1). During the decentralized execution, the duplex dueling component will be removed, and each agent will select actions using its individual Q-function based on local action-observation history.

**Individual Action-Value Function** is represented by a recurrent Q-network for each agent $i$, which takes previous hidden state $h_i^{t-1}$, current local observations $o_i^t$, and previous action $a_i^{t-1}$ as inputs and outputs local $Q_i(\tau_i,a_i)$.

**Duplex Dueling** component connects local and joint action-value functions via two modules: (i) a *Transformation* network module that incorporates the information of global state or joint history into individual action-value functions during the centralized training process, and (ii) a *Dueling Mixing* network module that composes separate action-value functions from *Transformation* into a joint action-value function. *Duplex Dueling* first derives the individual dueling structure for each agent $i$ by computing its value function $V_i(\tau_i) = \max_{a_i} Q_i(\tau_i, a_i)$ and its advantage function $A_i(\tau_i, a_i) = Q_i(\tau_i, a_i) - V_i(\tau_i)$, and then computes the joint dueling structure by using individual dueling structures.

**Transformation** network module uses the centralized information to transform local dueling structure $[V_i(\tau_i), A_i(\tau_i, a_i)]_{i=1}^n$ to $[V_i(\boldsymbol{\tau}), A_i(\boldsymbol{\tau}, a_i)]_{i=1}^n$ conditioned on the joint action-observation history, as shown below, for any agent $i$, i.e., $Q_i(\boldsymbol{\tau}, a_i) = w_i(\boldsymbol{\tau})Q_i(\tau_i, a_i) + b_i(\boldsymbol{\tau})$, thus,

$$V_i(\boldsymbol{\tau}) = w_i(\boldsymbol{\tau})V_i(\tau_i) + b_i(\boldsymbol{\tau}), \quad \text{and} \quad A_i(\boldsymbol{\tau}, a_i) = Q_i(\boldsymbol{\tau}, a_i) - V_i(\boldsymbol{\tau}) = w_i(\boldsymbol{\tau})A_i(\tau_i, a_i), \quad (7)$$

where $w_i(\boldsymbol{\tau}) > 0$ is a positive weight. This positive linear transformation maintains the consistency of the greedy action selection and alleviates partial observability in Dec-POMDP (Son et al., 2019; Yang et al., 2020). As used by QMIX (Rashid et al., 2018), QTRAN (Son et al., 2019), and Qatten (Yang et al., 2020), the centralized information can be the global state $s$, if available, or the joint action-observation history $\boldsymbol{\tau}$.

**Dueling Mixing** network module takes the outputs of the transformation network as input, e.g., $[V_i, A_i]_{i=1}^n$, and produces the values of joint $Q_{tot}$, as shown in Figure 1a. This dueling mixing network uses individual dueling structure transformed by *Transformation* to compute the joint value $V_{tot}(\boldsymbol{\tau})$ and the joint advantage $A_{tot}(\boldsymbol{\tau}, \boldsymbol{a})$, respectively, and finally outputs $Q_{tot}(\boldsymbol{\tau}, \boldsymbol{a}) = V_{tot}(\boldsymbol{\tau}) + A_{tot}(\boldsymbol{\tau}, \boldsymbol{a})$ by using the joint dueling structure.

Based on Fact 1, the advantage-based IGM principle imposes no constraints on value functions. Therefore, to enable efficient learning, we use a simple sum structure to compose the joint value:

$$V_{tot}(\boldsymbol{\tau}) = \sum_{i=1}^n V_i(\boldsymbol{\tau}) \qquad (8)$$

To enforce the IGM consistency of the joint advantage and individual advantages, as specified by Eq. (6), QPLEX computes the joint advantage function as follows:

$$A_{tot}(\boldsymbol{\tau}, \boldsymbol{a}) = \sum_{i=1}^n \lambda_i(\boldsymbol{\tau}, \boldsymbol{a})A_i(\boldsymbol{\tau}, a_i), \text{ where } \lambda_i(\boldsymbol{\tau}, \boldsymbol{a}) > 0. \qquad (9)$$

The joint advantage function $A_{tot}$ is the dot product of advantage functions $[A_i]_{i=1}^t$ and positive importance weights $[\lambda_i]_{i=1}^n$ with joint history and action. The positivity induced by $\lambda_i$ will continue to maintain the consistency flow of the greedy action selection and the joint information of $\lambda_i$ provides the full expressiveness power for value factorization. To enable efficient learning of importance weights $\lambda_i$ with joint history and action, QPLEX uses a scalable multi-head attention module (Vaswani et al., 2017):

$$\lambda_i(\boldsymbol{\tau}, \boldsymbol{a}) = \sum_{k=1}^K \lambda_{i,k}(\boldsymbol{\tau}, \boldsymbol{a})\phi_{i,k}(\boldsymbol{\tau})\upsilon_k(\boldsymbol{\tau}), \qquad (10)$$

where $K$ is the number of attention heads, $\lambda_{i,k}(\boldsymbol{\tau}, \boldsymbol{a})$ and $\phi_{i,k}(\boldsymbol{\tau})$ are attention weights activated by a sigmoid regularizer, and $\upsilon_k(\boldsymbol{\tau}) > 0$ is a positive key of each head. This sigmoid activation of $\lambda_i$ brings sparsity to the credit assignment of the joint advantage function to individuals, which enables efficient multi-agent learning (Wang et al., 2019b).

With Eq. (8) and (9), the joint action-value function $Q_{tot}$ can be reformulated as follows:

$$Q_{tot}(\boldsymbol{\tau}, \boldsymbol{a}) = V_{tot}(\boldsymbol{\tau}) + A_{tot}(\boldsymbol{\tau}, \boldsymbol{a}) = \sum_{i=1}^n Q_i(\boldsymbol{\tau}, a_i) + \sum_{i=1}^n \left(\lambda_i(\boldsymbol{\tau}, \boldsymbol{a}) - 1\right) A_i(\boldsymbol{\tau}, a_i). \qquad (11)$$

It can be seen that $Q_{tot}$ consists of two terms. The first term is the sum of action-value functions $[Q_i]_{i=1}^n$, which is the joint action-value function $Q_{tot}^{\text{Qatten}}$ of Qatten (Yang et al., 2020) (which is the $Q_{tot}$ of VDN (Sunehag et al., 2018) with global information). The second term corrects for the discrepancy between the centralized joint action-value function and $Q_{tot}^{\text{Qatten}}$, which is the main contribution of QPLEX to realize the full expressiveness power of value factorization.

| $a_1$ / $a_2$ | $\mathcal{A}^{(1)}$ | $\mathcal{A}^{(2)}$ | $\mathcal{A}^{(3)}$ |
|---|---|---|---|
| $\mathcal{A}^{(1)}$ | **8** | -12 | -12 |
| $\mathcal{A}^{(2)}$ | -12 | (∅) 6 | 0 |
| $\mathcal{A}^{(3)}$ | -12 | 0 | (∅) 6 |

(a) Payoff of a harder matrix game

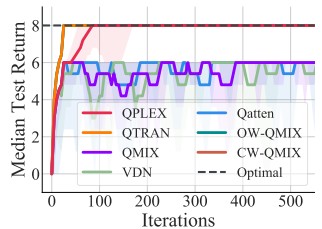

(b) Deep MARL algorithms

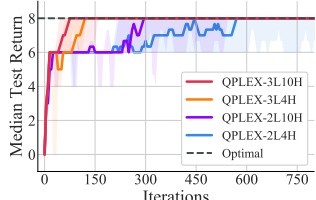

(c) Learning curves of ablation study

Figure 2: (a) Payoff matrix for a harder one-step game. Boldface means the optimal joint action selection from the payoff matrix. The strikethroughs indicate the original matrix game proposed by QTRAN. (b) The learning curves of QPLEX and other baselines. (c) The learning curve of QPLEX, whose suffix $a$L$b$H denotes the neural network size with $a$ layers and $b$ heads (multi-head attention) for learning importance weights $\lambda_i$ (see Eq. (9) and (10)), respectively.

**Proposition 2.** *Given the universal function approximation of neural networks, the action-value function class that QPLEX can realize is equivalent to what is induced by the IGM principle.*

In practice, QPLEX can utilize common neural network structures (e.g., multi-head attention modules) to achieve superior performance by approximating the universal approximation theorem (Csáji et al., 2001). We will discuss the effects of QPLEX's duplex dueling network with different configurations in Section 4.1. As introduced by Son et al. (2019) and Wang et al. (2020a), the completeness of value factorization is very critical for multi-agent Q-learning and we will illustrate the stability and state-of-the-art performance of QPLEX in online and offline data collections in the next section.

## 4 EXPERIMENTS

In this section, we first study didactic examples proposed by prior work (Son et al., 2019; Wang et al., 2020a) to investigate the effects of QPLEX's complete IGM expressiveness on learning optimality and stability. To demonstrate scalability on complex MARL domains, we also evaluate the performance of QPLEX on a range of StarCraft II benchmark tasks (Samvelyan et al., 2019). The completeness of the IGM function class can express richer joint action-value function classes induced by large and diverse datasets or training buffers. This expressiveness can provide QPLEX with higher sample efficiency to achieve state-of-the-art performance in online and offline data collections. We compare QPLEX with state-of-the-art baselines: QTRAN (Son et al., 2019), QMIX (Rashid et al., 2018), VDN (Sunehag et al., 2018), Qatten (Yang et al., 2020), and WQMIX (OW-QMIX and CW-QMIX; Rashid et al., 2020). In particular, the second term of Eq. (11) is the main difference between QPLEX and Qatten. Thus, Qatten provides a natural ablation baseline of QPLEX to demonstrate the effectiveness of this discrepancy term. The implementation details of these algorithms and experimental settings are deferred to Appendix B. We also conduct two ablation studies to study the influence of the attention structure of dueling architecture and the number of parameters on QPLEX, which are deferred to be discussed in Appendix E. Towards fair evaluation, all experimental results are illustrated with median performance and 25-75% percentiles over 6 random seeds.

### 4.1 MATRIX GAMES

QTRAN (Son et al., 2019) proposes a hard matrix game, as shown in Table 4a of Appendix C. In this subsection, we consider a harder matrix game in Table 2a, which also describes a simple cooperative multi-agent task with considerable miscoordination penalties, and its local optimum is more difficult to jump out. The optimal joint strategy of these two games is to perform action $\mathcal{A}^{(1)}$ simultaneously. To ensure sufficient data collection in the joint action space, we adopt uniform data distribution. With this fixed dataset, we can study the optimality of multi-agent Q-learning from an optimization perspective, ignoring the challenge of exploration and sample complexity.

As shown in Figure 2b, QPLEX, QTRAN, and WQMIX, which possess a richer expressiveness power of value factorization can achieve optimal performance, while other algorithms with limited expressiveness (e.g., QMIX, VDN, and Qatten) fall into a local optimum induced by miscoordination penalties. In the original matrix proposed by QTRAN, QPLEX and QTRAN can also successfully converge to optimal joint action-value functions. These results are deferred to Appendix C. QTRAN

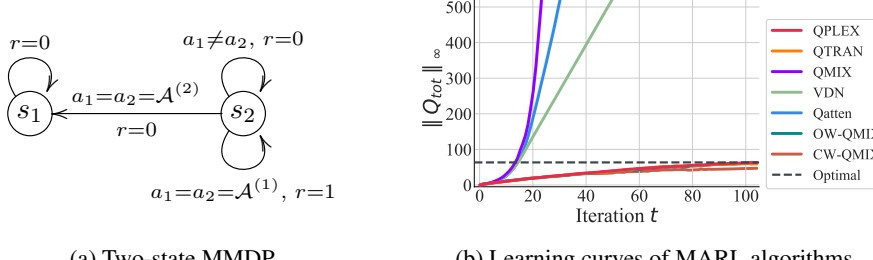

(a) Two-state MMDP     (b) Learning curves of MARL algorithms

Figure 3: (a) A special two-state MMDP used to demonstrate the training stability of the multi-agent Q-learning algorithms. $r$ is a shorthand for $r(s, \boldsymbol{a})$. (b) The learning curves of $\|Q_{tot}\|_{\infty}$ in a specific two-state MMDP.

achieves superior performance in the matrix games but suffers from its relaxation of IGM consistency in complex domains (such as StarCraft II) shown in Section 4.3.

In the theoretical analysis of QPLEX, Proposition 2 exploits the universal function approximation of neural networks. QPLEX allows the scalable implementations with various neural network capacities (different layers and heads of attention module) for learning importance weights $\lambda_i$ (see Eq. (9) and (10)). As shown in Figure 2c, by increasing the neural network size for learning $\lambda_i$ (e.g., QPLEX-3L10H), QPLEX possesses more expressiveness of value factorization and converges faster. However, learning efficiency becomes challenging for complex neural networks. To effectively perform StarCraft II tasks ranging from 2 to 27 agents, we use a small multi-head attention module (i.e., QPLEX-1L4H) in complex domains (see Section 4.3). Please refer to Appendix B for more detailed configurations.

## 4.2 Two-state MMDP

In this subsection, we focus on a Multi-agent Markov Decision Process (MMDP) (Boutilier, 1996) which is a fully cooperative multi-agent setting with full observability. Consider a two-state MMDP proposed by Wang et al. (2020a) with two agents, two actions, and a single reward (see Figure 3a). Two agents start at state $s_2$ and explore extrinsic rewards for 100 environment steps. The optimal policy of this MMDP is simply executing the action $\mathcal{A}^{(1)}$ at state $s_2$, which is the only coordination pattern to obtain the positive reward. To approximate the uniform data distribution, we adopt a uniform exploration strategy (i.e., $\epsilon$-greedy exploration with $\epsilon = 1$). We consider the training stability of multi-agent Q-learning algorithms with uniform data distribution in this special MMDP task. As shown in Figure 3b, the joint state-value function $Q_{tot}$ learned by baseline algorithms using limited function classes, including QMIX, VDN, and Qatten, will diverge. This instability phenomenon of VDN has been theoretically investigated by Wang et al. (2020a). By utilizing richer function classes, QPLEX, QTRAN, and WQMIX can address this numerical instability issue and converge to the optimal joint state-value function.

## 4.3 Decentralized StarCraft II Micromanagement Benchmark

A more challenging set of empirical experiments are based on StarCraft Multi-Agent Challenge (SMAC) benchmark (Samvelyan et al., 2019). We first investigate empirical performance in a popular experimental setting with $\epsilon$-greedy exploration and a limited first-in-first-out (FIFO) buffer (Samvelyan et al., 2019), named online data collection setting. To demonstrate the offline training potential of QPLEX, we also adopt the offline data collection setting proposed by Levine et al. (2020), which can be granted access to a given dataset without additional online exploration.

### 4.3.1 Training with Online Data Collection

We evaluate QPLEX in 17 benchmark tasks of StarCraft II, which contains 14 popular tasks proposed by SMAC (Samvelyan et al., 2019) and three new super hard cooperative tasks. To demonstrate the overall performance of each algorithm, Figure 4 plots the averaged median test win rate across all 17 scenarios and the number of scenarios in which the algorithm outperforms, respectively. Figure 4a shows that, compared with other baselines, QPLEX constantly and significantly outperforms baselines

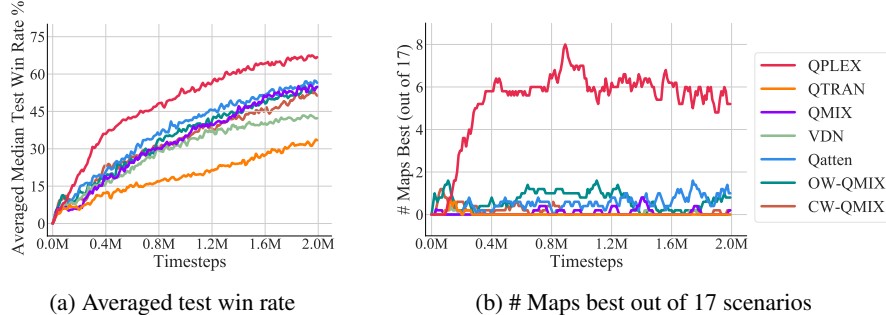

(a) Averaged test win rate          (b) # Maps best out of 17 scenarios

Figure 4: (a) The median test win %, averaged across all 17 scenarios. (b) The number of scenarios in which the algorithms' median test win % is the highest by at least 1/32 (smoothed).

over the whole training process and exceeds at least 10% median test win rate averaged across all 17 scenarios. Moreover, Figure 4b illustrates that, among all 17 tasks, QPLEX is the best performer on up to eight tasks, underperforms on just two tasks, and ties for the best performer on the rest tasks. After 0.8M timesteps, the number of tasks that QPLEX achieves the best performance gradually decreases to five, because, in several easy tasks, other baselines also reach almost 100% test win rate as shown in Figure 5. The overall evaluation diagram of the original SMAC benchmark (14 tasks) corresponding to Figure 4 is deferred to Figure 8 in Appendix D.

Figure 5 shows the learning curves on nine tasks in the online data collection setting and the results of other eight maps are deferred to Figure 7 in Appendix D. From Figure 5, we can observe that QPLEX significantly outperforms other baselines with higher sample efficiency. On the super hard map 5s10z, the performance gap between QPLEX and other baselines exceeds 30% in test win rate, and the visualized strategies of QPLEX and QMIX in this map are deferred to Appendix F. Most multi-agent Q-learning baselines including QMIX, VDN, and Qatten achieve reasonable performance (see Figure 5). However, as Figure 4 suggests, QTRAN performs the worst in these comparative experiments, even though it performs well in the didactic games. From a theoretical perspective, the online data collection process utilizes an $\epsilon$-greedy exploration process, which requires individual greedy action selections to build an effective training buffer. QTRAN may suffer from its relaxation of IGM consistency (soft constraints of IGM) in the online data collection phase, while the duplex dueling architecture of QPLEX (hard constraint of IGM) provides effective individual greedy action selections, making it suitable for data collection with $\epsilon$-greedy exploration.

Moreover, although WQMIX (OW-QMIX and CW-QMIX) outperforms QMIX in some tasks (illustrated in Figure 5, e.g., 2c_vs_64zg and bane_vs_bane), WQMIX show very similar overall performance as QMIX across 17 StarCraft II benchmark tasks. In contrast, QPLEX achieves significant improvement in convergence performance in a lot of hard and super hard maps and demonstrates high sample efficiency across most scenarios (see Figure 5).

### 4.3.2 TRAINING WITH OFFLINE DATA COLLECTION

Recently, offline reinforcement learning has been regarded as a key step for real-world RL applications (Dulac-Arnold et al., 2019; Levine et al., 2020). Agarwal et al. (2020) presents an optimistic perspective of offline Q-learning that DQN and its variants can achieve superior performance in Atari 2600 games (Bellemare et al., 2013) with sufficiently large and diverse datasets. In MARL, StarCraft II benchmark has the same discrete action space as Atari. We conduct a lot of experiments on the StarCraft II benchmark tasks to study offline multi-agent Q-learning in this subsection. We adopt a large and diverse dataset to make the expressiveness power of value factorization become the dominant factor to investigate. We train a behavior policy of QMIX and collect all its experienced transitions throughout the training process (see the details in Appendix C). As shown in Figure 13 in Appendix G, QPLEX significantly outperforms other multi-agent Q-learning baselines and possesses the state-of-the-art value factorization structure for offline multi-agent Q-learning. QMIX and Qatten cannot always maintain stable learning performance, and VDN suffers from offline data collection and leads to weak empirical results. QTRAN may perform well in certain cases when its soft constraints, two $\ell_2$-penalty terms, are well minimized. With offline data collection, individual greedy action selections do not need to build a training buffer, but they still need to compute the one-step TD target

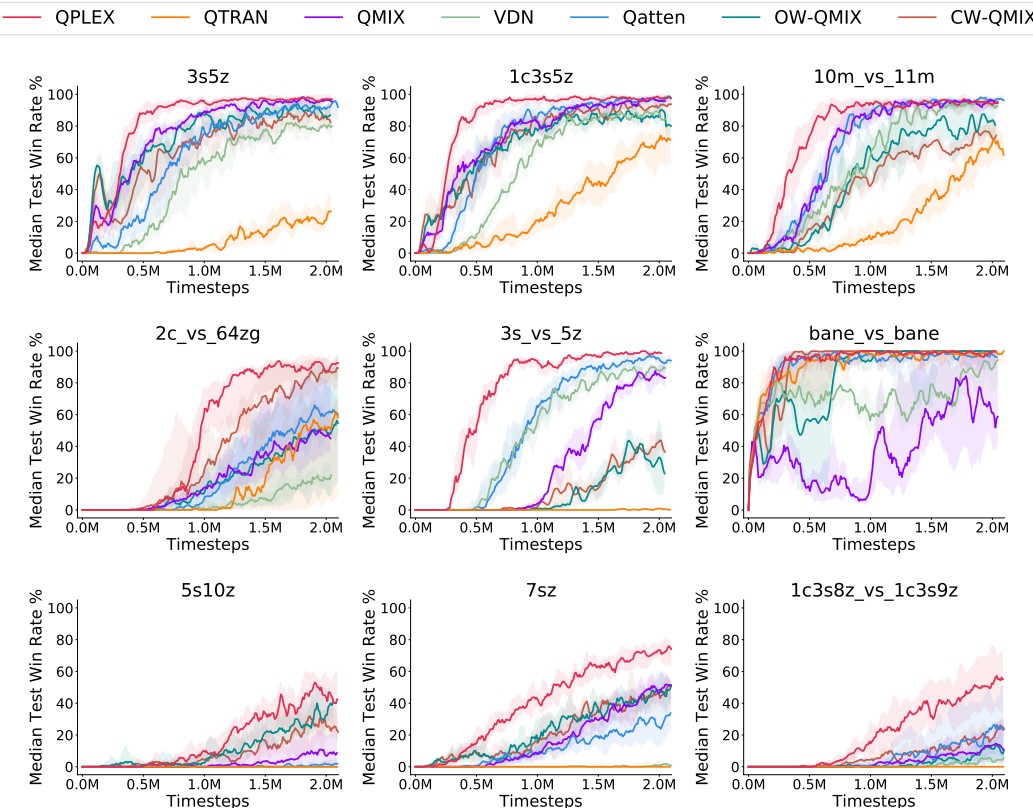

Figure 5: Learning curves of StarCraft II with online data collection.

for centralized training. Therefore, compared with QTRAN, QPLEX still has theoretical advantages regarding the IGM principle in the offline data collection setting.

## 5 CONCLUSION

In this paper, we introduced QPLEX, a novel multi-agent Q-learning framework that allows centralized end-to-end training and learns to factorize a joint action-value function to enable decentralized execution. QPLEX takes advantage of a duplex dueling architecture that efficiently encodes the IGM consistency constraint on joint and individual greedy action selections. Our theoretical analysis shows that QPLEX achieves a complete IGM function class. Empirical results demonstrate that it significantly outperforms state-of-the-art baselines in both online and offline data collection settings. In particular, QPLEX possesses strong ability of supporting offline training. This ability provides QPLEX with high sample efficiency and opportunities of utilizing offline multi-source datasets. It will be an interesting and valuable direction to study offline multi-agent reinforcement learning in continuous action spaces (such as MuJoCo (Todorov et al., 2012)) with QPLEX's value factorization.

## ACKNOWLEDGEMENTS

We would like to thank the anonymous reviewers for their insightful comments and helpful suggestions. This work is supported in part by Science and Technology Innovation 2030 – "New Generation Artificial Intelligence" Major Project (No. 2018AAA0100904), and a grant from the Institute of Guo Qiang, Tsinghua University.

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

# A  OMITTED PROOFS IN SECTION 3

**Definition 1** (Advantage-based IGM). *For a joint action-value function $Q_{tot}: \mathcal{T} \times \mathcal{A} \mapsto \mathbb{R}$ and individual action-value functions $[Q_i : \mathcal{T} \times \mathcal{A} \mapsto \mathbb{R}]_{i=1}^n$, where $\forall \boldsymbol{\tau} \in \mathcal{T}$, $\forall \boldsymbol{a} \in \mathcal{A}$, $\forall i \in \mathcal{N}$,*

$$\textit{(Joint Dueling)} \quad Q_{tot}(\boldsymbol{\tau}, \boldsymbol{a}) = V_{tot}(\boldsymbol{\tau}) + A_{tot}(\boldsymbol{\tau}, \boldsymbol{a}) \text{ and } V_{tot}(\boldsymbol{\tau}) = \max_{\boldsymbol{a'}} Q_{tot}(\boldsymbol{\tau}, \boldsymbol{a'}), \quad (3)$$

$$\textit{(Individual Dueling)} \quad Q_i(\tau_i, a_i) = V_i(\tau_i) + A_i(\tau_i, a_i) \text{ and } V_i(\tau_i) = \max_{a_i'} Q_i(\tau_i, a_i'), \quad (4)$$

*such that the following holds*

$$\arg\max_{\boldsymbol{a} \in \mathcal{A}} A_{tot}(\boldsymbol{\tau}, \boldsymbol{a}) = \left( \arg\max_{a_1 \in \mathcal{A}} A_1(\tau_1, a_1), \ldots, \arg\max_{a_n \in \mathcal{A}} A_n(\tau_n, a_n) \right), \quad (5)$$

*then, we can say that $[Q_i]_{i=1}^n$ satisfies advantage-based IGM for $Q_{tot}$.*

Let the action-value function class derived from IGM is denoted by

$$\widetilde{\mathcal{Q}} = \left\{ \left( \widetilde{Q}_{tot} \in \mathbb{R}^{|\mathcal{T}||\mathcal{A}|^n}, \left[ \widetilde{Q}_i \in \mathbb{R}^{|\mathcal{T}||\mathcal{A}|} \right]_{i=1}^n \right) \,\Big|\, \text{Eq. (2) is satisfied} \right\},$$

where $\widetilde{Q}_{tot}$ and $\left[ \widetilde{Q}_i \right]_{i=1}^n$ denote the joint and individual action-value functions induced by IGM, respectively. Similarly, let

$$\widehat{\mathcal{Q}} = \left\{ \left( \widehat{Q}_{tot} \in \mathbb{R}^{|\mathcal{T}||\mathcal{A}|^n}, \left[ \widehat{Q}_i \in \mathbb{R}^{|\mathcal{T}||\mathcal{A}|} \right]_{i=1}^n \right) \,\Big|\, \text{Eq. (3), (4), (5) are satisfied} \right\}$$

denote the action-value function class derived from advantage-based IGM. $\widehat{V}_{tot}$ and $\widehat{A}_{tot}$ denote the joint state-value and advantage functions, respectively. $\left[ \widehat{V}_i \right]_{i=1}^n$ and $\left[ \widehat{A}_i \right]_{i=1}^n$ denote the individual state-value and advantage functions induced by advantage-IGM, respectively. According to the duplex dueling architecture $Q = V + A$ stated in advantage-based IGM (see Definition 1), we derive the joint and individual action-value functions as following: $\forall \boldsymbol{\tau} \in \mathcal{T}$, $\forall \boldsymbol{a} \in \mathcal{A}$, $\forall i \in \mathcal{N}$,

$$\widehat{Q}_{tot}(\boldsymbol{\tau}, \boldsymbol{a}) = \widehat{V}_{tot}(\boldsymbol{\tau}) + \widehat{A}_{tot}(\boldsymbol{\tau}, \boldsymbol{a}) \quad \text{and} \quad \widehat{Q}_i(\tau_i, a_i) = \widehat{V}_i(\tau_i) + \widehat{A}_i(\tau_i, a_i).$$

**Proposition 1.** *The advantage-based IGM and IGM function classes are equivalent.*

*Proof.* We will prove $\widetilde{\mathcal{Q}} \equiv \widehat{\mathcal{Q}}$ in the following two directions.

$\widetilde{\mathcal{Q}} \subseteq \widehat{\mathcal{Q}}$  For any $\left( \widetilde{Q}_{tot}, \left[ \widetilde{Q}_i \right]_{i=1}^n \right) \in \widetilde{\mathcal{Q}}$, we construct $\widehat{Q}_{tot} = \widetilde{Q}_{tot}$ and $\left[ \widehat{Q}_i \right]_{i=1}^n = \left[ \widetilde{Q}_i \right]_{i=1}^n$. The joint and individual state-value/advantage functions induced by advantage-IGM

$$\widehat{V}_{tot}(\boldsymbol{\tau}) = \max_{\boldsymbol{a'}} \widehat{Q}_{tot}(\boldsymbol{\tau}, \boldsymbol{a'}) \quad \text{and} \quad \widehat{A}_{tot}(\boldsymbol{\tau}, \boldsymbol{a}) = \widehat{Q}_{tot}(\boldsymbol{\tau}, \boldsymbol{a}) - \widehat{V}_{tot}(\boldsymbol{\tau}),$$

$$\widehat{V}_i(\tau_i) = \max_{a_i'} \widehat{Q}_i(\tau_i, a_i') \quad \text{and} \quad \widehat{A}_i(\tau_i, a_i) = \widehat{Q}_i(\tau_i, a_i') - \widehat{V}_i(\tau_i), \quad \forall i \in \mathcal{N},$$

are derived by Eq. (3) and Eq. (4), respectively. Because state-value functions do not affect the greedy action selection, $\forall \boldsymbol{\tau} \in \mathcal{T}$, $\forall \boldsymbol{a} \in \mathcal{A}$,

$$\arg\max_{\boldsymbol{a} \in \mathcal{A}} \widetilde{Q}_{tot}(\boldsymbol{\tau}, \boldsymbol{a}) = \left( \arg\max_{a_1 \in \mathcal{A}} \widetilde{Q}_1(\tau_1, a_1), \ldots, \arg\max_{a_n \in \mathcal{A}} \widetilde{Q}_n(\tau_n, a_n) \right)$$

$$\Rightarrow \arg\max_{\boldsymbol{a} \in \mathcal{A}} \widehat{Q}_{tot}(\boldsymbol{\tau}, \boldsymbol{a}) = \left( \arg\max_{a_1 \in \mathcal{A}} \widehat{Q}_1(\tau_1, a_1), \ldots, \arg\max_{a_n \in \mathcal{A}} \widehat{Q}_n(\tau_n, a_n) \right)$$

$$\Rightarrow \arg\max_{\boldsymbol{a} \in \mathcal{A}} \left( \widehat{Q}_{tot}(\boldsymbol{\tau}, \boldsymbol{a}) - \widehat{V}_{tot}(\boldsymbol{\tau}) \right) =$$

$$\left( \arg\max_{a_1 \in \mathcal{A}} \left( \widehat{Q}_1(\tau_1, a_1) - \widehat{V}_1(\tau_1) \right), \ldots, \arg\max_{a_n \in \mathcal{A}} \left( \widehat{Q}_n(\tau_n, a_n) - \widehat{V}_n(\tau_n) \right) \right)$$

$$\Rightarrow \arg\max_{\boldsymbol{a} \in \mathcal{A}} \widehat{A}_{tot}(\boldsymbol{\tau}, \boldsymbol{a}) = \left( \arg\max_{a_1 \in \mathcal{A}} \widehat{A}_1(\tau_1, a_1), \ldots, \arg\max_{a_n \in \mathcal{A}} \widehat{A}_n(\tau_n, a_n) \right).$$

Thus, $\left( \widehat{Q}_{tot}, \left[ \widehat{Q}_i \right]_{i=1}^n \right) \in \widehat{\mathcal{Q}}$, which means that $\widetilde{\mathcal{Q}} \subseteq \widehat{\mathcal{Q}}$.

$\widehat{\mathcal{Q}} \subseteq \widetilde{\mathcal{Q}}$   We will prove this direction in the same way. For any $\left(\widehat{Q}_{tot}, \left[\widehat{Q}_i\right]_{i=1}^n\right) \in \widehat{\mathcal{Q}}$, we construct $\widetilde{Q}_{tot} = \widehat{Q}_{tot}$ and $\left[\widetilde{Q}_i\right]_{i=1}^n = \left[\widehat{Q}_i\right]_{i=1}^n$. Because state-value functions do not affect the greedy action selection, $\forall \boldsymbol{\tau} \in \boldsymbol{\mathcal{T}}, \forall \boldsymbol{a} \in \boldsymbol{\mathcal{A}}$,

$$\arg\max_{\boldsymbol{a} \in \boldsymbol{\mathcal{A}}} \widehat{A}_{tot}(\boldsymbol{\tau}, \boldsymbol{a}) = \left(\arg\max_{a_1 \in \mathcal{A}} \widehat{A}_1(\tau_1, a_1), \dots, \arg\max_{a_n \in \mathcal{A}} \widehat{A}_n(\tau_n, a_n)\right)$$

$$\Rightarrow \arg\max_{\boldsymbol{a} \in \boldsymbol{\mathcal{A}}} \left(\widehat{A}_{tot}(\boldsymbol{\tau}, \boldsymbol{a}) + \widehat{V}_{tot}(\boldsymbol{\tau})\right) =$$

$$\left(\arg\max_{a_1 \in \mathcal{A}} \left(\widehat{A}_1(\tau_1, a_1) + \widehat{V}_1(\tau_1)\right), \dots, \arg\max_{a_n \in \mathcal{A}} \left(\widehat{A}_n(\tau_n, a_n) + \widehat{V}_n(\tau_n)\right)\right)$$

$$\Rightarrow \arg\max_{\boldsymbol{a} \in \boldsymbol{\mathcal{A}}} \widehat{Q}_{tot}(\boldsymbol{\tau}, \boldsymbol{a}) = \left(\arg\max_{a_1 \in \mathcal{A}} \widehat{Q}_1(\tau_1, a_1), \dots, \arg\max_{a_n \in \mathcal{A}} \widehat{Q}_n(\tau_n, a_n)\right)$$

$$\Rightarrow \arg\max_{\boldsymbol{a} \in \boldsymbol{\mathcal{A}}} \widetilde{Q}_{tot}(\boldsymbol{\tau}, \boldsymbol{a}) = \left(\arg\max_{a_1 \in \mathcal{A}} \widetilde{Q}_1(\tau_1, a_1), \dots, \arg\max_{a_n \in \mathcal{A}} \widetilde{Q}_n(\tau_n, a_n)\right).$$

Thus, $\left(\widetilde{Q}_{tot}, \left[\widetilde{Q}_i\right]_{i=1}^n\right) \in \widetilde{\mathcal{Q}}$, which means that $\widehat{\mathcal{Q}} \subseteq \widetilde{\mathcal{Q}}$. The action-value function classes derived from advantage-based IGM and IGM are equivalent. □

**Fact 1.** *The constraint of advantage-based IGM stated in Eq. (5) is equivalent to that when* $\forall \boldsymbol{\tau} \in \boldsymbol{\mathcal{T}}$, $\forall \boldsymbol{a}^* \in \boldsymbol{\mathcal{A}}^*(\boldsymbol{\tau}), \forall \boldsymbol{a} \in \boldsymbol{\mathcal{A}} \setminus \boldsymbol{\mathcal{A}}^*(\boldsymbol{\tau}), \forall i \in \mathcal{N}$,

$$A_{tot}(\boldsymbol{\tau}, \boldsymbol{a}^*) = A_i(\tau_i, a_i^*) = 0 \quad and \quad A_{tot}(\boldsymbol{\tau}, \boldsymbol{a}) < 0, A_i(\tau_i, a_i) \leq 0, \tag{6}$$

*where* $\boldsymbol{\mathcal{A}}^*(\boldsymbol{\tau}) = \{\boldsymbol{a} | \boldsymbol{a} \in \boldsymbol{\mathcal{A}}, Q_{tot}(\boldsymbol{\tau}, \boldsymbol{a}) = V_{tot}(\boldsymbol{\tau})\}$.

*Proof.* We derive that $\forall \boldsymbol{\tau} \in \boldsymbol{\mathcal{T}}, \forall \boldsymbol{a} \in \boldsymbol{\mathcal{A}}, \forall i \in \mathcal{N}, \widehat{A}_{tot}(\boldsymbol{\tau}, \boldsymbol{a}) \leq 0$ and $\widehat{A}_i(\tau_i, a_i) \leq 0$ from Eq. (3) and Eq. (4) of Definition 1, respectively. According to the definition of $\arg\max$ operator, Eq. (3), and Eq. (4), $\forall \boldsymbol{\tau} \in \boldsymbol{\mathcal{T}}$, let $\widehat{\boldsymbol{\mathcal{A}}}^*(\boldsymbol{\tau})$ denote $\arg\max_{\boldsymbol{a} \in \boldsymbol{\mathcal{A}}} \widehat{A}_{tot}(\boldsymbol{\tau}, \boldsymbol{a})$ as follows:

$$\widehat{\boldsymbol{\mathcal{A}}}^*(\boldsymbol{\tau}) = \arg\max_{\boldsymbol{a} \in \boldsymbol{\mathcal{A}}} \widehat{A}_{tot}(\boldsymbol{\tau}, \boldsymbol{a}) = \arg\max_{\boldsymbol{a} \in \boldsymbol{\mathcal{A}}} \widehat{Q}_{tot}(\boldsymbol{\tau}, \boldsymbol{a})$$

$$= \left\{\boldsymbol{a} | \boldsymbol{a} \in \boldsymbol{\mathcal{A}}, \widehat{Q}_{tot}(\boldsymbol{\tau}, \boldsymbol{a}) = \widehat{V}_{tot}(\boldsymbol{\tau})\right\}$$

$$= \left\{\boldsymbol{a} | \boldsymbol{a} \in \boldsymbol{\mathcal{A}}, \widehat{Q}_{tot}(\boldsymbol{\tau}, \boldsymbol{a}) - \widehat{V}_{tot}(\boldsymbol{\tau}) = 0\right\}$$

$$= \left\{\boldsymbol{a} | \boldsymbol{a} \in \boldsymbol{\mathcal{A}}, \widehat{A}_{tot}(\boldsymbol{\tau}, \boldsymbol{a}) = 0\right\}. \tag{12}$$

Similarly, $\forall \boldsymbol{\tau} \in \boldsymbol{\mathcal{T}}, \forall i \in \mathcal{N}$, let $\widehat{\mathcal{A}}_i^*(\tau_i)$ denote $\arg\max_{a_i \in \mathcal{A}} \widehat{A}_i(\tau_i, a_i)$ as follows:

$$\widehat{\mathcal{A}}_i^*(\tau_i) = \arg\max_{a_i \in \mathcal{A}} \widehat{A}_i(\tau_i, a_i) = \arg\max_{a_i \in \mathcal{A}} \widehat{Q}_i(\tau_i, a_i)$$

$$= \left\{a_i | a_i \in \mathcal{A}, \widehat{Q}_i(\tau_i, a_i) = \widehat{V}_i(\tau_i)\right\}$$

$$= \left\{a_i | a_i \in \mathcal{A}, \widehat{A}_i(\tau_i, a_i) = 0\right\}. \tag{13}$$

Thus, $\forall \boldsymbol{\tau} \in \boldsymbol{\mathcal{T}}, \forall \boldsymbol{a}^* \in \widehat{\boldsymbol{\mathcal{A}}}^*(\boldsymbol{\tau}), \forall \boldsymbol{a} \in \boldsymbol{\mathcal{A}} \setminus \widehat{\boldsymbol{\mathcal{A}}}^*(\boldsymbol{\tau})$,

$$\widehat{A}_{tot}(\boldsymbol{\tau}, \boldsymbol{a}^*) = 0 \quad and \quad \widehat{A}_{tot}(\boldsymbol{\tau}, \boldsymbol{a}) < 0; \tag{14}$$

$\forall \boldsymbol{\tau} \in \boldsymbol{\mathcal{T}}, \forall i \in \mathcal{N}, \forall a_i^* \in \widehat{\mathcal{A}}^*(\tau_i), \forall a_i \in \mathcal{A} \setminus \widehat{\mathcal{A}}^*(\tau_i)$,

$$\widehat{A}_i(\tau_i, a_i^*) = 0 \quad and \quad \widehat{A}_i(\tau_i, a_i) < 0. \tag{15}$$

Recall the constraint stated in Eq. (5), $\forall \boldsymbol{\tau} \in \boldsymbol{\mathcal{T}}$,

$$\arg\max_{\boldsymbol{a} \in \boldsymbol{\mathcal{A}}} \widehat{A}_{tot}(\boldsymbol{\tau}, \boldsymbol{a}) = \left(\arg\max_{a_1 \in \mathcal{A}} \widehat{A}_1(\tau_1, a_1), \dots, \arg\max_{a_n \in \mathcal{A}} \widehat{A}_n(\tau_n, a_n)\right).$$

We can rewrite the constraint of advantage-based IGM stated in Eq. (5) as $\forall \boldsymbol{\tau} \in \mathcal{T}$,

$$\widehat{\boldsymbol{\mathcal{A}}}^{*}(\boldsymbol{\tau}) = \left\{ \langle a_1, \ldots, a_n \rangle | a_i \in \widehat{\mathcal{A}}_i^{*}(\tau_i), \forall i \in \mathcal{N} \right\}. \tag{16}$$

Therefore, combining Eq. (14), Eq. (15), and Eq. (16), we can derive $\forall \boldsymbol{\tau} \in \mathcal{T}$, $\forall \boldsymbol{a}^{*} \in \widehat{\boldsymbol{\mathcal{A}}}^{*}(\boldsymbol{\tau})$, $\forall \boldsymbol{a} \in \boldsymbol{\mathcal{A}} \setminus \widehat{\boldsymbol{\mathcal{A}}}^{*}(\boldsymbol{\tau}), \forall i \in \mathcal{N}$,

$$\widehat{A}_{tot}(\boldsymbol{\tau}, \boldsymbol{a^*}) = \widehat{A}_i(\tau_i, a_i^*) = 0 \quad \text{and} \quad \widehat{A}_{tot}(\boldsymbol{\tau}, \boldsymbol{a}) < 0, \widehat{A}_i(\tau_i, a_i) \leq 0. \tag{17}$$

In another way, combining Eq. (14), Eq. (15), and Eq. (17), we can derive Eq. (16) by the definition of $\widehat{\boldsymbol{\mathcal{A}}}^{*}$ and $\left[\widehat{\mathcal{A}}^{*}\right]_{i=1}^{n}$ (see Eq. (12) and Eq. (13)). In more detail, the closed set property of Cartesian product of $[a_i^*]_{i=1}^{n}$ has been encoded into the Eq. (16) and Eq. (17) simultaneously. □

**Proposition 2.** *Given the universal function approximation of neural networks, the action-value function class that QPLEX can realize is equivalent to what is induced by the IGM principle.*

*Proof.* We assume that the neural network of QPLEX can be large enough to achieve the universal function approximation by corresponding theorem (Csáji et al., 2001). Let the action-value function class that QPLEX can realize is denoted by

$$\overline{\mathcal{Q}} = \left\{ \left( \overline{Q}_{tot} \in \mathbb{R}^{|\mathcal{T}||\mathcal{A}|^n}, \left[\overline{Q}_i \in \mathbb{R}^{|\mathcal{T}||\mathcal{A}|}\right]_{i=1}^{n} \right) \middle| \text{Eq. (7), (8), (9), (10), (11) are satisfied} \right\}.$$

In addition, $\overline{Q}_{tot}, \overline{V}_{tot}, \overline{A}_{tot}, \left[\overline{Q}'_i\right]_{i=1}^{n}, \left[\overline{V}'_i\right]_{i=1}^{n}, \left[\overline{A}'_i\right]_{i=1}^{n}, \left[\overline{Q}_i\right]_{i=1}^{n}, \left[\overline{V}_i\right]_{i=1}^{n}$, and $\left[\overline{A}_i\right]_{i=1}^{n}$ denote the corresponding (joint, transformed, and individual) (action-value, state-value, and advantage) functions, respectively. In the implementation of QPLEX, we ensure the positivity of important weights of *Transformation* and joint advantage function, $[w_i]_{i=1}^{n}$ and $[\lambda_i]_{i=1}^{n}$, which maintains the greedy action selection flow and rules out these non-interesting points (zeros) on optimization. We will prove $\widehat{\mathcal{Q}} \equiv \overline{\mathcal{Q}}$ in the following two directions.

$\widehat{\mathcal{Q}} \subseteq \overline{\mathcal{Q}}$ For any $\left( \widehat{Q}_{tot}, \left[\widehat{Q}_i\right]_{i=1}^{n} \right) \in \widehat{\mathcal{Q}}$, we construct $\overline{Q}_{tot} = \widehat{Q}_{tot}$ and $\left[\overline{Q}_i\right]_{i=1}^{n} = \left[\widehat{Q}_i\right]_{i=1}^{n}$ and derive $\overline{V}_{tot}, \overline{A}_{tot}, \left[\overline{V}_i\right]_{i=1}^{n}$, and $\left[\overline{A}_i\right]_{i=1}^{n}$ by Eq.(3) and Eq. (4), respectively. Note that in the construction of QPLEX,

$$V_i(\boldsymbol{\tau}) = w_i(\boldsymbol{\tau})V_i(\tau_i) + b_i(\boldsymbol{\tau}) \quad \text{and} \quad A_i(\boldsymbol{\tau}, a_i) = w_i(\boldsymbol{\tau})A_i(\tau_i, a_i)$$

and

$$Q_{tot}(\boldsymbol{\tau}, \boldsymbol{a}) = V_{tot}(\boldsymbol{\tau}) + A_{tot}(\boldsymbol{\tau}, \boldsymbol{a}) = \sum_{i=1}^{n} V_i(\boldsymbol{\tau}) + \sum_{i=1}^{n} \lambda_i(\boldsymbol{\tau}, \boldsymbol{a})A_i(\boldsymbol{\tau}, a_i).$$

In addition, we construct transformed functions connecting joint and individual functions as follows: $\forall \boldsymbol{\tau} \in \mathcal{T}, \forall \boldsymbol{a} \in \boldsymbol{\mathcal{A}}, \forall i \in \mathcal{N}$,

$$\overline{Q}'_i(\boldsymbol{\tau}, \boldsymbol{a}) = \frac{\overline{Q}_{tot}(\boldsymbol{\tau}, \boldsymbol{a})}{n}, \ \overline{V}'_i(\boldsymbol{\tau}) = \arg\max_{\boldsymbol{a} \in \boldsymbol{\mathcal{A}}} \overline{Q}'_i(\boldsymbol{\tau}, \boldsymbol{a}), \text{ and } \overline{A}'_i(\boldsymbol{\tau}, \boldsymbol{a}) = \overline{Q}'_i(\boldsymbol{\tau}, \boldsymbol{a}) - \overline{V}'_i(\boldsymbol{\tau}),$$

which means that according to Fact 1,

$$w_i(\boldsymbol{\tau}) = 1, \ b_i(\boldsymbol{\tau}) = \overline{V}'_i(\boldsymbol{\tau}) - \overline{V}_i(\tau_i), \text{ and } \lambda_i(\tau_i, \boldsymbol{a}) = \begin{cases} \dfrac{\overline{A}'_i(\tau_i, \boldsymbol{a})}{\overline{A}_i(\tau_i, a_i)} > 0, & \text{when } \overline{A}_i(\tau_i, a_i) < 0, \\ 1, & \text{when } \overline{A}_i(\tau_i, a_i) = 0. \end{cases}$$

Thus, $\left( \overline{Q}_{tot}, \left[\overline{Q}_i\right]_{i=1}^{n} \right) \in \overline{\mathcal{Q}}$, which means that $\widehat{\mathcal{Q}} \subseteq \overline{\mathcal{Q}}$.

$\overline{\mathcal{Q}} \subseteq \widehat{\mathcal{Q}}$   For any $\left( \overline{Q}_{tot}, [\overline{Q}_i]_{i=1}^n \right) \in \overline{\mathcal{Q}}$, with the similar discussion of Fact 1, $\forall \boldsymbol{\tau} \in \mathcal{T}, \forall i \in \mathcal{N}$, let $\overline{\mathcal{A}}_i^*(\tau_i)$ denote $\arg \max_{a_i \in \mathcal{A}} \overline{A}_i(\tau_i, a_i)$, where

$$\overline{\mathcal{A}}_i^*(\tau_i) = \left\{ a_i | a_i \in \mathcal{A}, \overline{A}_i(\tau_i, a_i) = 0 \right\}.$$

Combining the positivity of $[w_i]_{i=1}^n$ and $[\lambda_i]_{i=1}^n$ with Eq. (7), (8), (9), and (11), we can derive $\forall \boldsymbol{\tau} \in \mathcal{T}, \forall i \in \mathcal{N}, \forall a_i^* \in \overline{\mathcal{A}}^*(\tau_i), \forall a_i \in \mathcal{A} \setminus \overline{\mathcal{A}}^*(\tau_i)$,

$$\overline{A}_i(\tau_i, a_i^*) = 0 \quad \text{and} \quad \overline{A}_i(\tau_i, a_i) < 0$$
$$\Rightarrow \quad \overline{A}_i'(\boldsymbol{\tau}, a_i^*) = w_i(\boldsymbol{\tau}) \overline{A}_i(\tau_i, a_i^*) = 0 \quad \text{and} \quad \overline{A}_i'(\boldsymbol{\tau}, a_i) = w_i(\boldsymbol{\tau}) \overline{A}_i(\tau_i, a_i) < 0$$
$$\Rightarrow \quad \overline{A}_{tot}(\boldsymbol{\tau}, \boldsymbol{a}^*) = \lambda_i(\boldsymbol{\tau}, \boldsymbol{a}^*) \overline{A}_i'(\tau_i, a_i^*) = 0 \quad \text{and} \quad \overline{A}_{tot}(\boldsymbol{\tau}, \boldsymbol{a}) = \lambda_i(\boldsymbol{\tau}, \boldsymbol{a}) \overline{A}_i'(\tau_i, a_i^*) < 0,$$

where $\boldsymbol{a}^* = \langle a_1^*, \ldots, a_n^* \rangle$ and $\boldsymbol{a} = \langle a_1, \ldots, a_n \rangle$. Notably, these $\boldsymbol{a}^*$ forms

$$\overline{\boldsymbol{\mathcal{A}}}^*(\boldsymbol{\tau}) = \left\{ \langle a_1, \ldots, a_n \rangle | a_i \in \overline{\mathcal{A}}_i^*(\tau_i), \forall i \in \mathcal{N} \right\}, \tag{18}$$

which is similar to Eq. (16) in the proof of Fact 1. We construct $\widehat{Q}_{tot} = \overline{Q}_{tot}$ and $\left[ \widehat{Q}_i \right]_{i=1}^n = \left[ \overline{Q}_i \right]_{i=1}^n$. According to Eq. (18), the constraints of advantage-based IGM stated in Fact 1 (Eq. (3), Eq. (4), and Eq. (6)) are satisfied, which means that $\left( \widehat{Q}_{tot}, \left[ \widehat{Q}_i \right]_{i=1}^n \right) \in \widehat{\mathcal{Q}}$ and $\overline{\mathcal{Q}} \subseteq \widehat{\mathcal{Q}}$.

Thus, when assuming neural networks provide universal function approximation, the joint action-value function class that QPLEX can realize is equivalent to what is induced by the IGM principle.   □

## B   EXPERIMENT SETTINGS AND IMPLEMENTATION DETAILS

### B.1   STARCRAFT II

We consider the combat scenario of StarCraft II unit micromanagement tasks, where the enemy units are controlled by the built-in AI, and each ally unit is controlled by the reinforcement learning agent. The units of the two groups can be asymmetric, but the units of each group should belong to the same race. At each timestep, every agent takes action from the discrete action space, which includes the following actions: no-op, move [direction], attack [enemy id], and stop. Under the control of these actions, agents move and attack in continuous maps. At each time step, MARL agents will get a global reward equal to the total damage done to enemy units. Killing each enemy unit and winning the combat will bring additional bonuses of 10 and 200, respectively. We briefly introduce the SMAC challenges of our paper in Table 1.

### B.2   IMPLEMENTATION DETAILS

We adopt the PyMARL (Samvelyan et al., 2019) implementation of state-of-the-art baselines: QTRAN (Son et al., 2019), QMIX (Rashid et al., 2018), VDN (Sunehag et al., 2018), Qatten (Yang et al., 2020), and WQMIX (OW-QMIX and CW-QMIX; Rashid et al., 2020). The hyper-parameters of these algorithms are the same as that in SMAC (Samvelyan et al., 2019) and referred in their source codes. QPLEX is also based on PyMARL, whose special hyper-parameters are illustrated in Table 2 and other common hyper-parameters are adopted by the default implementation of PyMARL (Samvelyan et al., 2019). Especially in the online data collection, we take the advanced implementation of *Transformation* of Qatten in QPLEX. To ensure the positivity of important weights of *Transformation* and joint advantage function, we add a sufficiently small amount $\epsilon' = 10^{-10}$ on $[w_i]_{i=1}^n$ and $[\lambda_i]_{i=1}^n$. In addition, we stop gradients of local advantage function $A_i$ to increase the optimization stability of the max operator of dueling structure. This instability consideration due to max operator has been justified by Dueling DQN (Wang et al., 2016). We approximate the joint action-value function as

$$Q_{tot}(\boldsymbol{\tau}, \boldsymbol{a}) \approx \sum_{i=1}^n Q_i(\boldsymbol{\tau}, a_i) + \sum_{i=1}^n \left( \lambda_i(\boldsymbol{\tau}, \boldsymbol{a}) - 1 \right) \widetilde{A}_i(\boldsymbol{\tau}, a_i),$$

where $\widetilde{A}_i$ denotes a variant of the local advantage function $A_i$ by stoping gradients.

| Map Name | Ally Units | Enemy Units |
|---|---|---|
| 2s3z | 2 Stalkers & 3 Zealots | 2 Stalkers & 3 Zealots |
| 3s5z | 3 Stalkers & 5 Zealots | 3 Stalkers & 5 Zealots |
| 1c3s5z | 1 Colossus, 3 Stalkers & 5 Zealots | 1 Colossus, 3 Stalkers & 5 Zealots |
| 5m_vs_6m | 5 Marines | 6 Marines |
| 10m_vs_11m | 10 Marines | 11 Marines |
| 27m_vs_30m | 27 Marines | 30 Marines |
| 3s5z_vs_3s6z | 3 Stalkers & 5 Zealots | 3 Stalkers & 6 Zealots |
| MMM2 | 1 Medivac, 2 Marauders & 7 Marines | 1 Medivac, 2 Marauders & 8 Marines |
| 2s_vs_1sc | 2 Stalkers | 1 Spine Crawler |
| 3s_vs_5z | 3 Stalkers | 5 Zealots |
| 6h_vs_8z | 6 Hydralisks | 8 Zealots |
| bane_vs_bane | 20 Zerglings & 4 Banelings | 20 Zerglings & 4 Banelings |
| 2c_vs_64zg | 2 Colossi | 64 Zerglings |
| corridor | 6 Zealots | 24 Zerglings |
| 5s10z | 5 Stalkers & 10 Zealots | 5 Stalkers & 10 Zealots |
| 7sz | 7 Stalkers & 7 Zealots | 7 Stalkers & 7 Zealots |
| 1c3s8z_vs_1c3s9z | 1 Colossus, 3 Stalkers & 8 Zealots | 1 Colossus, 3 Stalkers & 9 Zealots |

Table 1: The StarCraft multi-Agent challenge (SMAC; Samvelyan et al., 2019) benchmark.

| QPLEX's architecuture configurations | Didactic Examples | StarCraft II |
|---|---|---|
| The number of layers in $w, b, \lambda, \phi, \upsilon$ | 2 or 3 | 1 |
| The number of heads in the attention module | 4 or 10 | 4 |
| Unit number in middle layers of $w, b, \lambda, \phi, \upsilon$ | 64 | $\varnothing$ |
| Activation in the middle layers of $w, \upsilon$ | Relu | $\varnothing$ |
| Activation in the last layer of $w, \upsilon$ | Absolute | Absolute |
| Activation in the middle layers of $b$ | Relu | $\varnothing$ |
| Activation in the last layer of $b$ | None | None |
| Activation in the middle layers of $\lambda, \phi$ | Relu | $\varnothing$ |
| Activation in the last layer of $\lambda, \phi$ | Sigmoid | Sigmoid |

Table 2: The network configurations of QPLEX's architecture.

Our training time on an NVIDIA RTX 2080TI GPU of each task is about 6 hours to 20 hours, depending on the agent number and the episode length limit of each map. The percentage of episodes in which MARL agents defeat all enemy units within the time limit is called *test win rate*. We pause training every 10k timesteps and evaluate 32 episodes with decentralized greedy action selection to measure *test win rate* of each algorithm. After training every 200 episodes, the target network will be updated once. We call this update period an *Iteration* for didactic tasks. In the two-state MMDP, *Optimal* line of Figure 3b is approximately $\sum_{i=0}^{99} \gamma^i = 63.4$ in one episode of 100 timesteps.

**Training with Online Data Collection**   We have collected a total of 2 million timestep data for each task and test the model every 10 thousand steps. We use $\epsilon$-greedy exploration and a limited first-in-first-out (FIFO) replay buffer of size 5000 episodes, where $\epsilon$ is linearly annealed from 1.0 to 0.05 over 50k timesteps and keep it constant for the rest training process. To utilize the training buffer more efficiently, we perform gradient updates twice with a batch of 32 episodes after collecting every episode for each algorithm.

**Training with Offline Data Collection**   To construct a diverse dataset, we train a behavior policy of QMIX (Rashid et al., 2018) or VDN (Sunehag et al., 2018) and collect its 20k or 50k experienced episodes throughout the training process. The dataset configurations are shown in Table 3. We evaluate QPLEX and four baselines over six random seeds, which includes three different datasets and tests two seeds on each dataset. We train 300 epochs to demonstrate our learning performance, where each epoch trains 160k transitions with a batch of 32 episodes. Moreover, the training process of behavior policy is the same as that discussed in PyMARL (Samvelyan et al., 2019).

| Map Name | Replay Buffer Size | Behaviour Test Win Rate | Behaviour Policy |
|----------|-------------------|------------------------|------------------|
| 2s3z | 20k episodes | 95.8% | QMIX |
| 3s5z | 20k episodes | 92.0% | QMIX |
| 1c3s5z | 20k episodes | 90.2% | QMIX |
| 2s_vs_1sc | 20k episodes | 98.1% | QMIX |
| 3s_vs_5z | 20k episodes | 94.4% | VDN |
| 2c_vs_64zg | 50k episodes | 80.9% | QMIX |

Table 3: The dataset configurations of the offline data collection setting.

## C  OMITTED FIGURES AND TABLES IN SECTION 4.1 AND 4.2

| $a_1$ / $a_2$ | $\mathcal{A}^{(1)}$ | $\mathcal{A}^{(2)}$ | $\mathcal{A}^{(3)}$ |
|------|------|------|------|
| $\mathcal{A}^{(1)}$ | **8** | -12 | -12 |
| $\mathcal{A}^{(2)}$ | -12 | 0 | 0 |
| $\mathcal{A}^{(3)}$ | -12 | 0 | 0 |

(a) Payoff of matrix game

| $a_1$ / $a_2$ | $\mathcal{A}^{(1)}$ | $\mathcal{A}^{(2)}$ | $\mathcal{A}^{(3)}$ |
|------|------|------|------|
| $\mathcal{A}^{(1)}$ | **8.0** | -12.1 | -12.1 |
| $\mathcal{A}^{(2)}$ | -12.2 | -0.0 | -0.0 |
| $\mathcal{A}^{(3)}$ | -12.1 | -0.0 | -0.0 |

(b) $Q_{tot}$ of QPLEX

| $a_1$ / $a_2$ | $\mathcal{A}^{(1)}$ | $\mathcal{A}^{(2)}$ | $\mathcal{A}^{(3)}$ |
|------|------|------|------|
| $\mathcal{A}^{(1)}$ | **8.0** | -12.0 | -12.0 |
| $\mathcal{A}^{(2)}$ | -12.0 | -0.0 | 0.0 |
| $\mathcal{A}^{(3)}$ | -12.0 | 0.0 | 0.0 |

(c) $Q_{tot}$ of QTRAN

| $a_1$ / $a_2$ | $\mathcal{A}^{(1)}$ | $\mathcal{A}^{(2)}$ | $\mathcal{A}^{(3)}$ |
|------|------|------|------|
| $\mathcal{A}^{(1)}$ | -8.0 | -8.0 | -8.0 |
| $\mathcal{A}^{(2)}$ | -8.0 | -0.0 | **-0.0** |
| $\mathcal{A}^{(3)}$ | -8.0 | -0.0 | -0.0 |

(d) $Q_{tot}$ of QMIX

| $a_1$ / $a_2$ | $\mathcal{A}^{(1)}$ | $\mathcal{A}^{(2)}$ | $\mathcal{A}^{(3)}$ |
|------|------|------|------|
| $\mathcal{A}^{(1)}$ | -6.2 | -4.9 | -4.9 |
| $\mathcal{A}^{(2)}$ | -4.9 | -3.6 | -3.6 |
| $\mathcal{A}^{(3)}$ | -4.9 | **-3.6** | -3.6 |

(e) $Q_{tot}$ of VDN

| $a_1$ / $a_2$ | $\mathcal{A}^{(1)}$ | $\mathcal{A}^{(2)}$ | $\mathcal{A}^{(3)}$ |
|------|------|------|------|
| $\mathcal{A}^{(1)}$ | -6.2 | -4.9 | -4.9 |
| $\mathcal{A}^{(2)}$ | -4.9 | -3.5 | **-3.5** |
| $\mathcal{A}^{(3)}$ | -4.9 | -3.5 | -3.5 |

(f) $Q_{tot}$ of Qatten

Table 4: (a) Payoff matrix of the one-step game. Boldface means the optimal joint action selection from payoff matrix. (b-f) The joint action-value functions $Q_{tot}$ of QPLEX, QTRAN, QMIX, VDN, and Qatten. Boldface means greedy joint action selection from joint action-value functions.

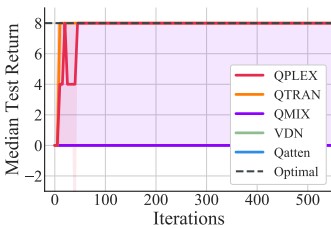

Figure 6: The learning curves of QPLEX and other baselines on the origin matrix game.

# D    EXPERIMENTS ON STARCRAFT II WITH ONLINE DATA COLLECTION

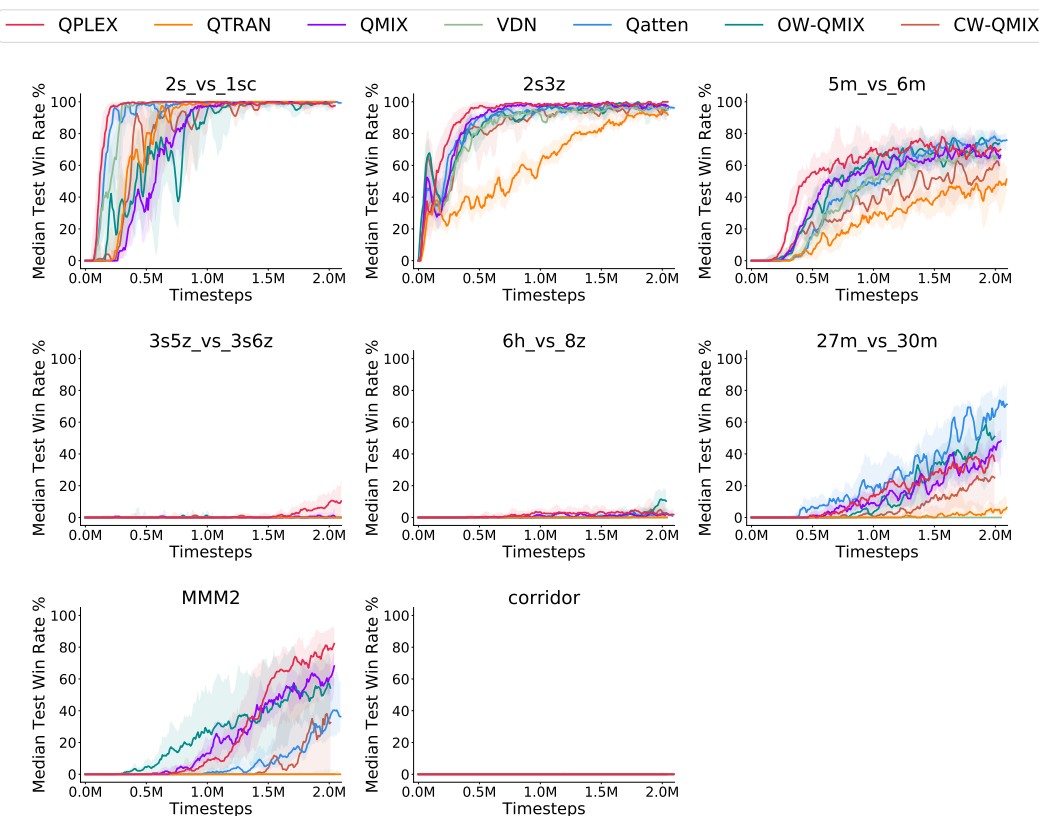

Figure 7: The learning curves of StarCraft II with online data collection on remaining scenarios.

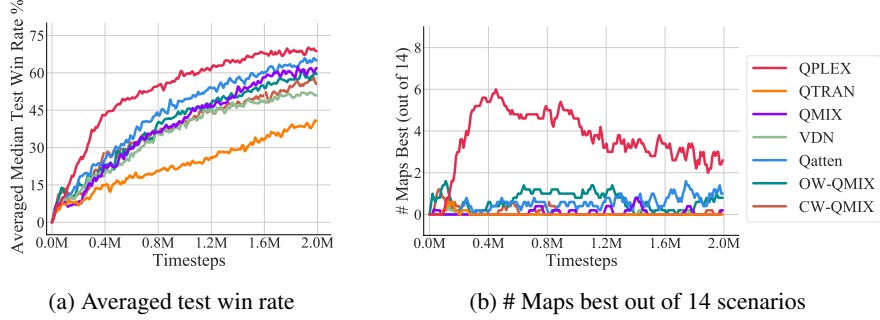

(a) Averaged test win rate          (b) # Maps best out of 14 scenarios

Figure 8: (a) The median test win %, averaged across all 14 scenarios proposed by SMAC (Samvelyan et al., 2019). (b) The number of scenarios in which the algorithms' median test win % is the highest by at least 1/32 (smoothed).

# E   ABLATION STUDIES WITH ONLINE DATA COLLECTION

In this section, we conduct two ablation studies to investigate why QPLEX works, which includes: (i) QPLEX without the multi-head attention structure of dueling architecture, and (ii) QMIX with the same number of parameters as QPLEX. For these studies, Figure 9 plots the averaged median test win rate % over the tasks of the StarCraft II benchmark mentioned in Section 4.3.1. Detailed learning curves on each task are shown in Figure 10 and 11, respectively.

Multi-head attention structure of importance weights $\lambda_i$ (see Eq. (9) and (10)) allows QPLEX to adapt its scalable implementation to different scenarios, e.g., didactic games or StarCraft II benchmark tasks. Section 4.1 demonstrates the importance of this attention structure, which can provide QPLEX more expressiveness of value factorization to perform better on the didactic matrix games. In this ablation study, we aim to test whether this multi-head attention is necessary for this StarCraft II domain. Specifically, we use a one-layer forward model instead of this multi-head attention structure in QPLEX, which is called *QPLEX-wo-duel-atten*. Figure 9a shows that QPLEX-wo-duel-atten achieves similar performance as QPLEX, which indicates that the the superiority of QPLEX over other MARL methods is largely due to the duplex dueling architecture (see Figure 1), instead of the multi-head attention trick.

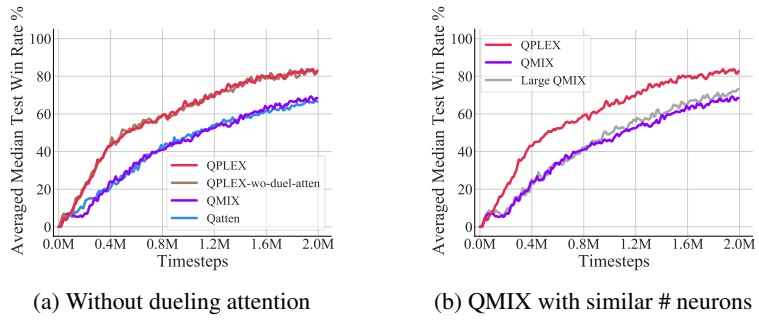

(a) Without dueling attention          (b) QMIX with similar # neurons

Figure 9: Ablation studies on QPLEX with the median test win %, averaged benchmark scenarios.

QPLEX uses more parameters in the value factorization architecture of neural networks due to its multi-head attention structure. We introduce *Large QMIX* with a similar number of parameters with QPLEX, to investigate whether the superiority of QPLEX over QMIX is due to the increase in the number of parameters. Figure 9b shows that QMIX with a larger network cannot fundamentally improve its performance, and QPLEX still significantly outperforms Large QMIX by a large margin. This result also confirms that the main effect of QPLEX comes from its novel value factorization structure (duplex dueling architecture) rather than the number of parameters.

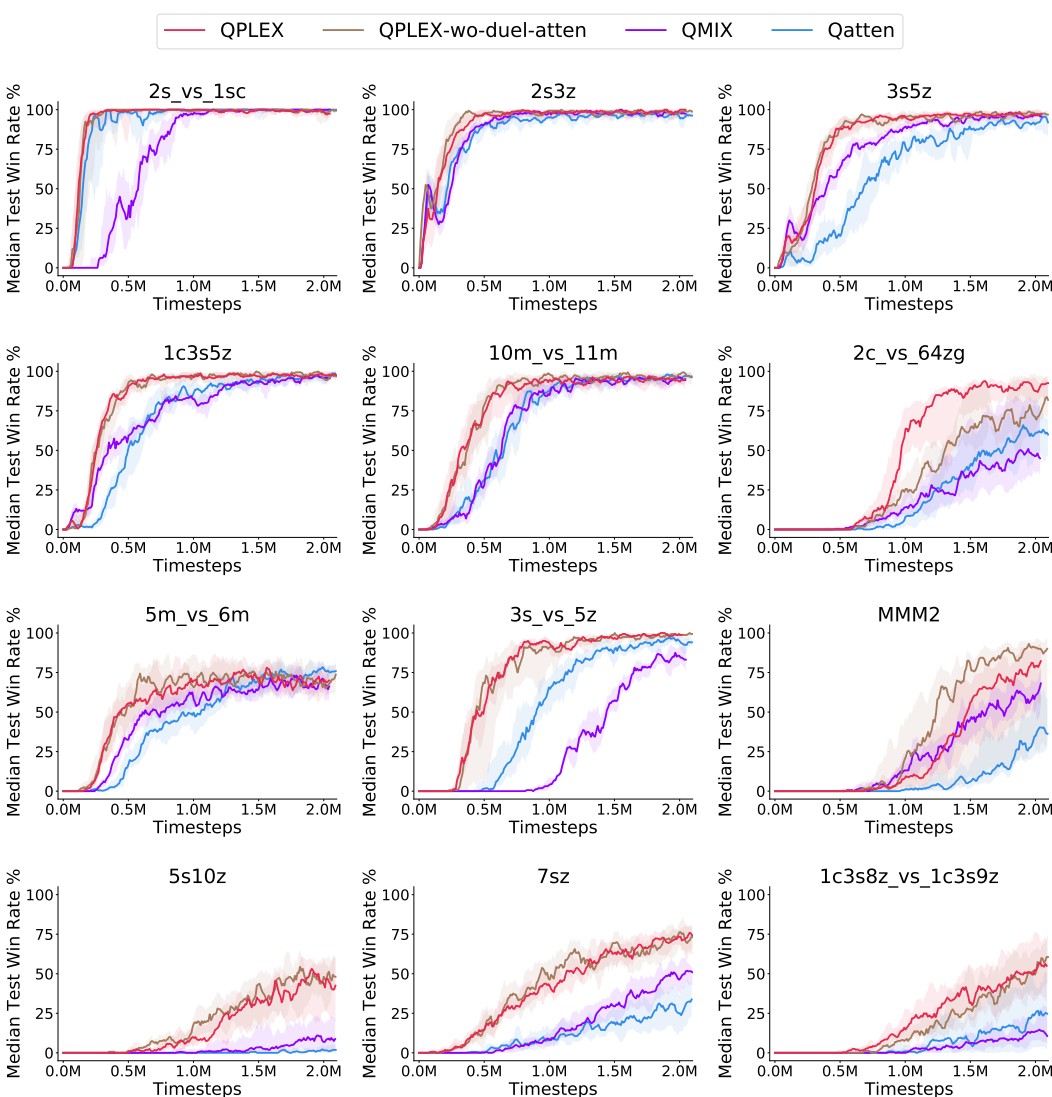

Figure 10: The learning curves of median test win rate % for QPLEX, QPLEX's ablation QPLEX-wo-duel-atten, QMIX, and Qatten with online data collection.

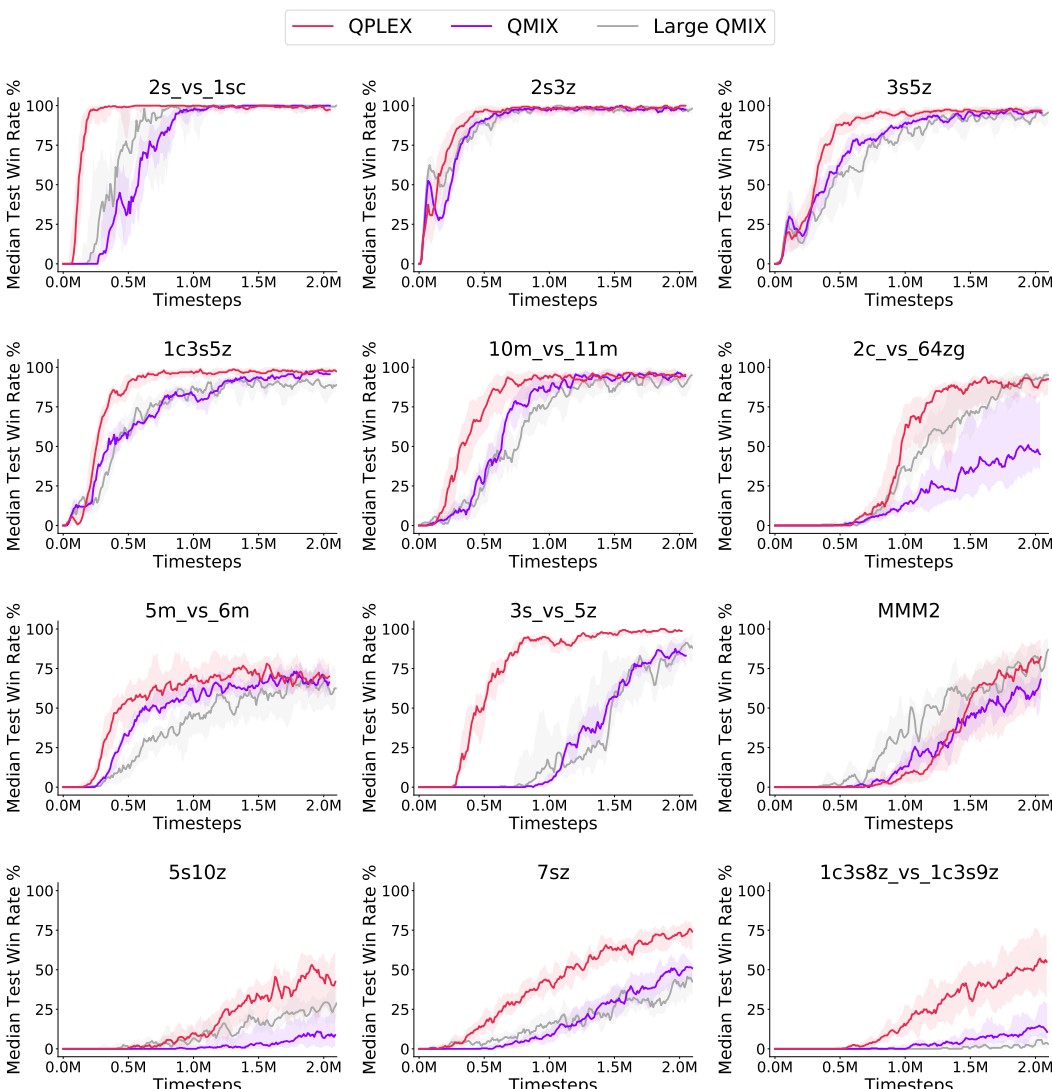

Figure 11: Learning curves of median test win rate % for QPLEX, QMIX, and Large QMIX with online data collection.

## F  A VISUALIZATION OF THE STRATEGIES LEARNED IN 5S10Z

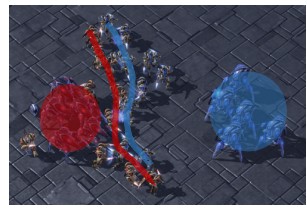

(a) Strategy of QPLEX on 5s10z                    (b) Strategy of QMIX on 5s10z

Figure 12: Visualized strategies of QPLEX and QMIX on 5s10z map of StarCraft II benchmark. Red marks represent learning agents, and blue marks represent build-in AI agents.

As shown in Figure 12, both MARL agents and opponents contain 5 ranged soldiers (denoted by a circle) and 10 melee soldiers (denoted by line) on 5s10z map. The ranged soldiers have stronger combat capabilities and need to be protected strategically. QPLEX uses 10 melee soldiers to build lines of defense against the enemy, while QMIX fails to coordinate melee soldiers such that ranged soldiers have to fight against the enemy directly.

## G  EXPERIMENTS ON STARCRAFT II WITH OFFLINE DATA COLLECTION

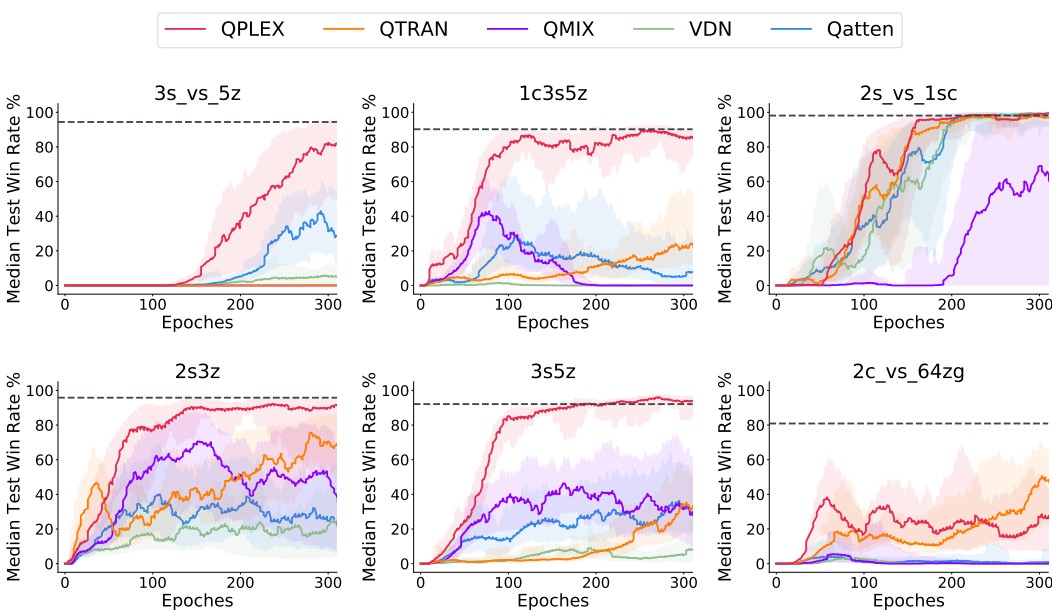

Figure 13: Deferred learning curves of StarCraft II with offline data collection on tested scenarios.

# H ABLATION STUDIES ABOUT QPLEX WITH DIFFERENT NETWORK CAPACITIES IN STARCRAFT II

We trade off the expressiveness and learning efficiency of the multi-head attention network module for estimating importance weights $\lambda_i$ (see Eq. (9) and (10) in Section 3.2). It is generally sufficient for QPLEX to use a simple multi-head attention with a small number of heads and layers (as evaluated in StarCraft II tasks) to achieve the state-of-the-art performance shown in Figure 4. However, in some didactic corner-cases with an adequate and uniform dataset, a harder matrix game illustrated in Figure 2a requires very high precision in estimating the action-value function in order to differentiate the optimal solution from the sub-optimal solutions. For this matrix game, a multi-head attention structure with more layers and heads has a more representational capacity and results in better performance, as demonstrated by the ablation study illustrated in Figure 2c in Section 4.1.

In contrast, StarCraft II micromanagement benchmark tasks contain much more complicated agents with large state-action spaces and range from 2 to 27 agents. To support the superior training scalability of QPLEX, we used a multi-head attention with just one layer and four heads. To evaluate the effect of the choice of layer and the number of heads, we conducted an ablation study in Starcraft II benchmark tasks. For simplicity, we follow the notation of Figure 2, i.e., use QPLEX-$a$L$b$H to denote QPLEX with $a$ layers and $b$ heads of importance weights $\lambda_i$. As shown in Figure 14, using more heads, QPLEX-1L10H, does not change the performance, but using more layers, QPLEX-2L4H, may slightly degenerate the learning efficiency of QPLEX in this complex domain. Detailed learning curves on these tasks are

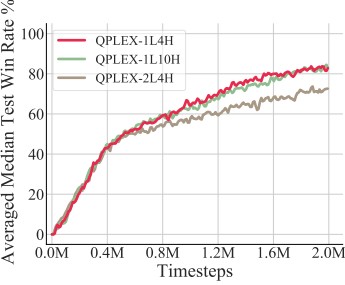

Figure 14: Ablation study about QPLEX with different network capacities in Star-Craft II.

shown in Figure 15. This is because using more layers significantly increase the number of parameters and requires more samples for learning, which may offset the benefits of higher expressiveness it brings. This ablation study shows that a simple attention neural network with just one layer and four heads has enough expressiveness to handle these complex StarCraft II tasks.

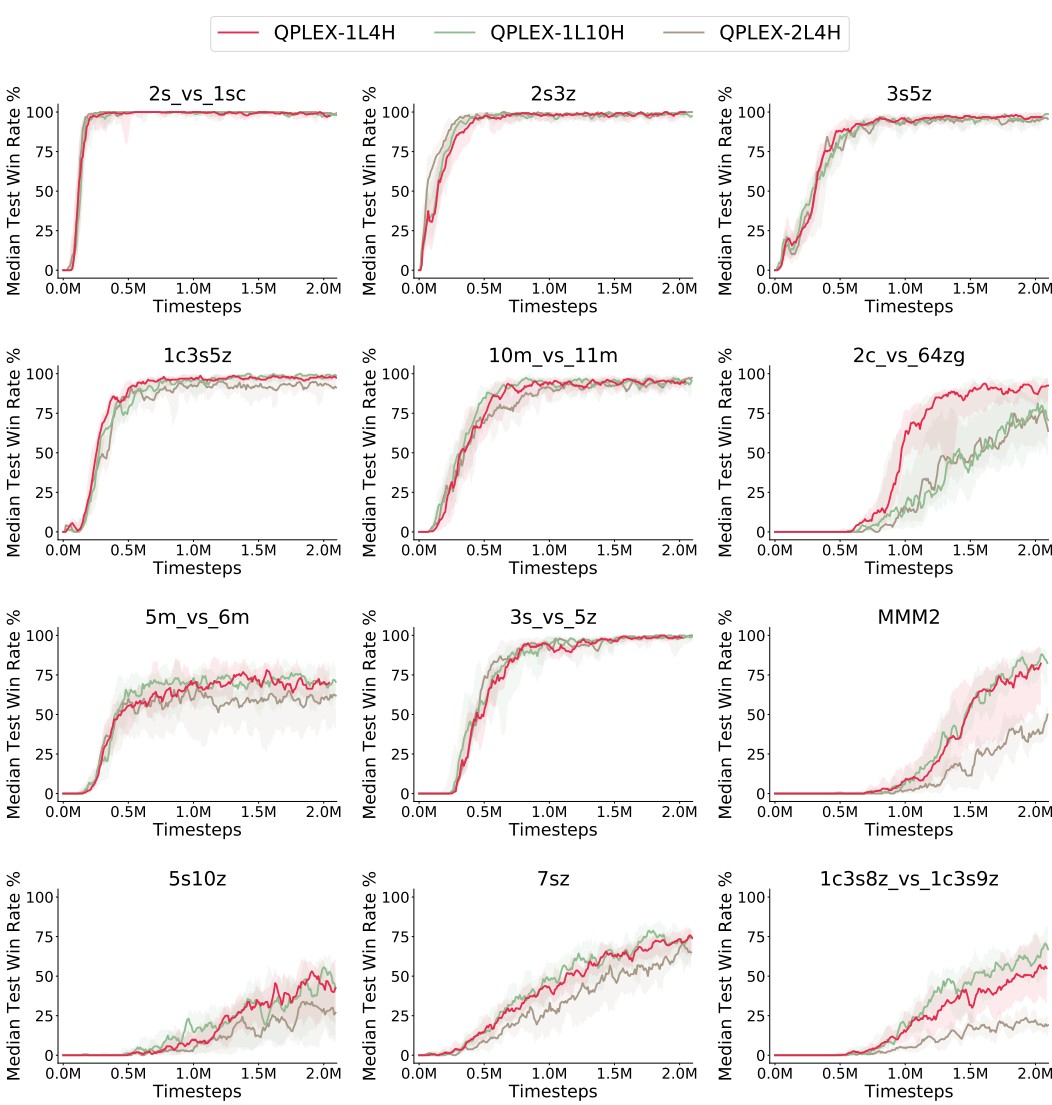

Figure 15: Deferred figures of median test win rate % for QPLEX-1L4H, QPLEX-1L10H, and QPLEX-2L4H with online data collection.

# I  ABLATION STUDIES ABOUT QTRAN

Both QPLEX and QTRAN aim to provide a richer factorized acton-value function class. The main difference between QPLEX and QTRAN is that QPLEX uses a duplex dueling architecture to realize the IGM principle (a hard constraint). In contrast, QTRAN uses two penalties as soft constraints to approximate the IGM principle. Moreover, from the perspective of implementation, QTRAN does not have a *Transformation* module (see Section 3.2) on the individual Q-functions and cannot utilize a multi-head attention module on the joint Q-function directly (because QTRAN does not take the duplex dueling architecture as QPLEX).

To test whether QPLEX outperforms QTRAN because of these factors, we conducted an ablation study by removing the *Transformation* module and replacing the multi-head attention module with a simple one-layer forward model in the QPLEX's dueling architecture, which is denoted as *QPLEX-wo-trans-atten*. In addition, we also introduce a variant of QTRAN, which also uses the *Transformation* module for individual Q-functions, denoted as *QTRAN-w-trans*. Our experiments are evaluated on the tasks of the StarCraft II benchmark mentioned in Section 4.3.1.

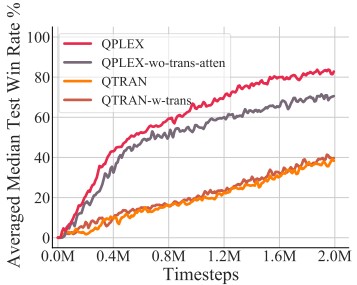

Figure 16: Ablation study about QTRAN with online data collection.

Figure 16 illustrates the averaged median test win rate % over all tested scenarios. Detailed learning curves on these tasks are shown in Figure 17. These empirical results show that QPLEX-wo-trans-atten significantly outperforms QTRAN and QTRAN-w-trans, which implies that the outperformance of QPLEX over QTRAN is largely due to its duplex dueling architecture. It can also be seen that QTRAN-w-trans cannot significantly improve the performance of QTRAN, which implies that QTRAN cannot benefit a lot from an extra *Transformation* module empirically.

As discussed in Section E, Figure 9a can be regarded as an ablation study of *QPLEX-wo-atten*, which just removes the multi-head attention module from dueling architecture. That ablation study shows that a simple neural network implementation of QPLEX's dueling structure has enough expressiveness to handle these StarCraft II tasks in the online data collection setting, even if this structure can provide QPLEX with excellent performance in didactic matrix games using an adequate and uniform dataset (see Section 4.1). Thus, compared with QPLEX-wo-atten in Figure 9a, Figure 16 shows that *Transformation* (abbreviated as *-trans*) is a useful module for QPLEX empirically, which indicates an another QPLEX's advantage, i.e., QPLEX can equip a *Transformation* module to improve QPLEX's empirical performance, whereas QTRAN may not by directly using it. Moreover, we would like to emphasize that, unlike the multi-head attention module, the *Transformation* module is actually a necessary module for QPLEX to realize IGM, as shown in Figure 1 and by the proof of Proposition 2. Therefore, it is actually fair to evaluate QPLEX with the *Transformation* module.

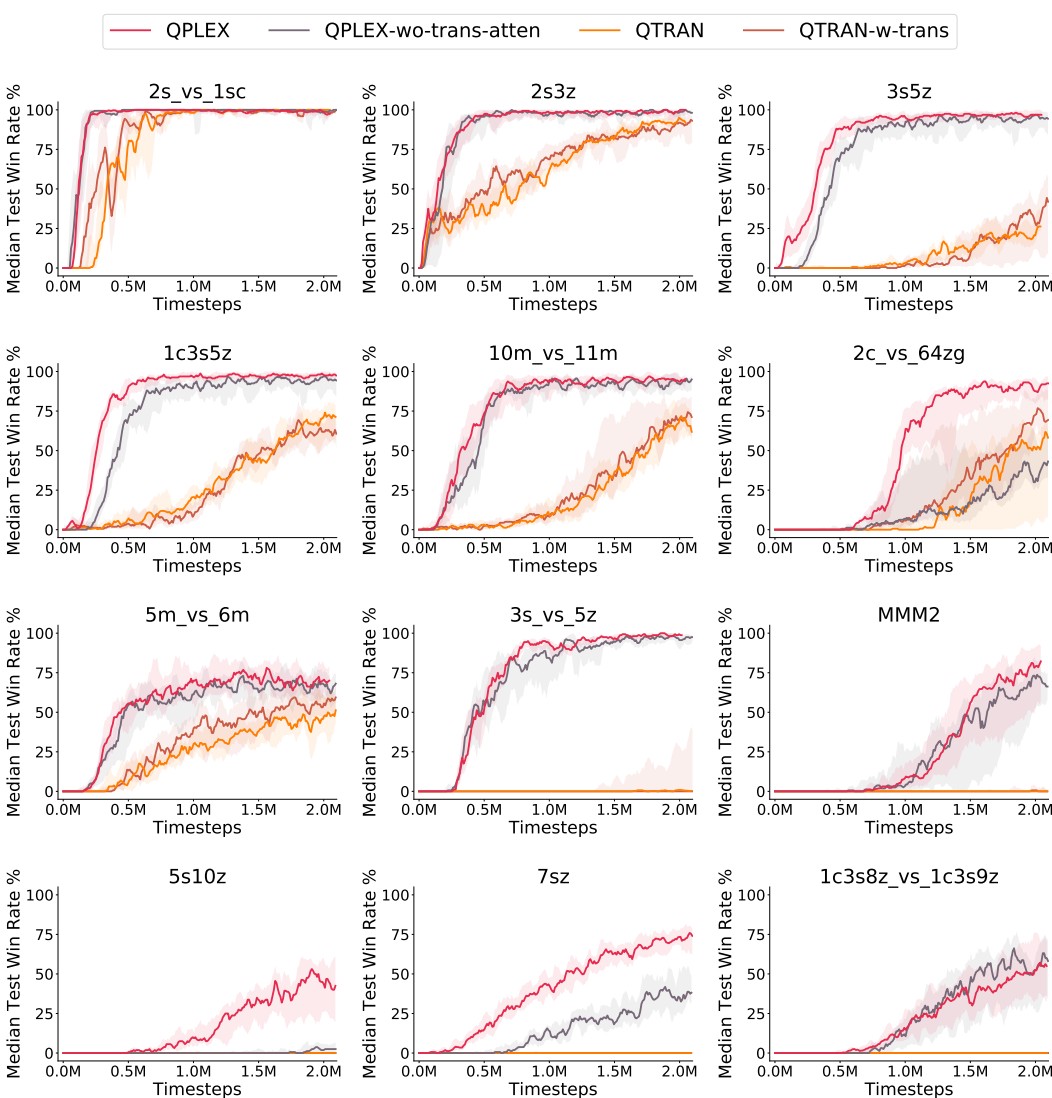

Figure 17: Figures of median test win rate % for QPLEX, QPLEX's ablation QPLEX-wo-trans-atten, QTRAN, and QTRAN-w-trans with online data collection.

## J A COMPARISON TO WQMIX IN PREDATOR-PREY

WQMIX (Rashid et al., 2020) is a recent advanced multi-agent Q-learning algorithm. We compare QPLEX with WQMIX in a toy game, *predator-prey*, which is introduced by WQMIX and aims to test coordination between agents in a partially-observable setting.

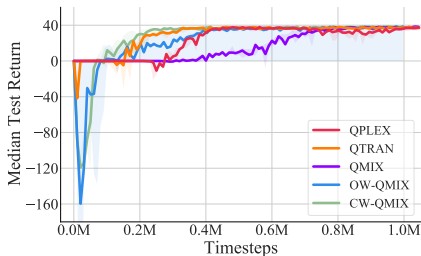

Figure 18: Learning curves of median test return for QPLEX, QTRAN, QMIX, and WQMIX (OW-QMIX and CW-QMIX) in the toy predator-prey task.

Predator-prey is a multi-agent coordinated game used by WQMIX (Rashid et al., 2020) with miscoordination penalties. In order to collect the experience with the positive reward of agents' coordinated actions, extensive exploration can benefit multi-agent Q-learning algorithms to solve this kind of tasks. WQMIX shapes the data distribution with an importance weight to boost efficient learning, which can also be regarded as a type of biased exploration. To support QPLEX with effective exploration, we use an $\epsilon$-greedy strategy which is also discussed in the paper of WQMIX. This strategy's $\epsilon$ is linearly annealed from 1 to 0.05 over 1 million timesteps, increased from the 50k used in SMAC (Samvelyan et al., 2019). As shown in Figure 18, besides WQMIX and QTRAN, QPLEX can solve this task by using the introduced $\epsilon$-greedy exploration strategy. Moreover, QMIX can also solve this task by using the same $\epsilon$-greedy strategy as QPLEX, but QPLEX enjoys higher sample efficiency.

