# OpenReview forum: "QPLEX: Duplex Dueling Multi-Agent Q-Learning"
_ICLR.cc/2021/Conference — ICLR 2021 Poster_

### Official Review · AnonReviewer2 · 2020-10-24

**Rating:** 4
**Confidence:** 4

**Review:**

This paper proposes a novel value decomposition approach to learn decentralized Q function in multi-agent setting. This idea is to follow Individual-Global-Max (IGM) principle. The main contribution is to use dueling structure (Q_i = V_i+A_i) for each agent i, and separately combining advantage/value terms respectively to form a centralized Q and A terms for training. Such combinations keep the positive correlation constraint between Q_tot and individual agent Q_i in QMix (i.e,. \partial Q_{tot} / \partial Q_i > 0) via positive trainable module in neural network, implemented by multi-head attention, etc.

While this paper seems to be a reasonable contribution, a bunch of critical issues exist in the experiments so I vote for rejection.

1. The idea is quite similar/incremental to QMIX, but instead of a mixing network, the authors use two-stages (first a linear weight that depends on all states, then a linear mixture over all agents with attention) to enforce the positive correlation how an individual Q would affect Q_tot. I wonder what’s the key component that makes QPLEX much better than QMIX? Is that purely because you have more parameters in the mixing? There don’t seem to be ablation studies (the video has sub-captions called “ablation study” but no sentence on what’s going on in the two videos) and discussions.

2. Qatten uses attention while the remaining baselines (e.g., QMix, VDN, QTRAN) do not have attention. Is QPLEX working because of attention rather than the dueling structure? How is QPLEX compared with a version of itself without attention?

3. In StarCraft Multi-agent Challenge, 5s10z is neither a hard or a super hard map. The hard maps are 2c_vs_64zg, bane_vs_bane, 5m_vs_6m and 3s_vs_5z, and the super hard maps are MMM2, corridor, 3s5z_vs_3s6z, 6h_vs_8z and 27m_vs_30m (See Tbl. 1, Fig. 5 and Fig. 6 in SMAC paper). I wonder why Fig. 4 and Fig. 5 are tested on different set of maps and how does QPLEX perform in hard/super hard maps (e.g., no comparison for online setting in 2c_vs_64zg, and for offline setting, QPLEX doesn’t see to perform well on 2c_vs_64zg). Note that the Qatten paper shows performance on these hard/super hard tasks.

Minor:
1. What does the strikethrough mean in Fig. 2a (pay off matrix)?

---

> ### Author Response · Authors · 2020-11-22
> **Response to Reviewer 2 (Part 1)**
>
> Thanks for your comments. We provide clarification to your questions and concerns as below. We appreciate if you have any further questions or comments.
>
> **Q1**: The idea is quite similar/incremental to QMIX. I wonder what’s the key component that makes QPLEX much better than QMIX?
>
> From the perspective of value factorization, QPLEX is fundamentally different from QMIX. The major advantage of QPLEX over QMIX is its superior representational capacity of the factorized joint value function class. Although QMIX uses a nonlinear combination of local Q-functions, QMIX can only realizes the monotonic value functions, which is just a proper subset of the complete joint value function class satisfying the IGM principle. Because of its limited representation capacity, (MAVEN; Mahajan et al., 2019) theoretically reveals the non-optimality of QMIX's value factorization in a class of tasks. In contrast, QPLEX uses a duplex dueling architecture and can realize the complete value function class induced by the IGM principle (see Proposition 1, Fact 1, and Proposition 2), which enables its outperformance over QMIX.
>
> **Q2**: Is that purely because you have more parameters in the mixing?
>
> This is not true. We conduct an ablation study by introducing *Large QMIX* with more neurons and a similar number of parameters with QPLEX. As illustrated in Figure 6b in Section 4.3.2 of the new revision, empirical results over Starcraft benchmark tasks show that QMIX with a larger network has similar performance as the original QMIX and QPLEX still significantly outperforms Large QMIX by a large margin. This result also confirms that the main effect of QPLEX comes from its novel value factorization structure (duplex dueling architecture) rather than the number of parameters. Detailed learning curves on tested tasks are shown in Appendix F.
>
> **Q3**: Is QPLEX working because of attention rather than the dueling structure? How is QPLEX compared with a version of itself without attention?
>
> Ablation study shows that QPLEX's outperformance in the StarCraft II domain is not because of the multi-head attention in the dueling architecture, but the dueling structure itself. In Section 4.3.2 of the new revision, we include an extra ablation study that evaluates QPLEX with a one-layer forward model instead of the multi-head attention structure, denoted as *QPLEX-wo-duel-atten*. As illustrated in Figure 6a with performance averaged over Starcraft benchmark tasks, empirical results show that QPLEX-wo-duel-atten achieves similar performance as the original QPLEX, which indicates that the superiority of QPLEX over other MARL methods is largely due to the duplex dueling architecture (see Figure 1), instead of the multi-head attention trick. Detailed learning curves on tested tasks are shown in Appendix E.
>
> **Q4**: I wonder why Fig. 4 and Fig. 5 are tested on different set of maps and how does QPLEX perform in hard/super hard maps, e.g., no comparison for online setting in 2c\_vs\_64zg.
>
> We have also conducted extensive experiments in more tasks and included these results in Section 4.3 of the new revision. We evaluate QPLEX in 17 scenarios of the StarCraft II benchmark, which contains 14 popular standard maps proposed by SMAC (Samvelyan et al., 2019) and three new super hard cooperative tasks. With this thorough evaluation, QPLEX demonstrates significant outperformance over other baselines, as shown by Figure 4, 5, and 8.
>
> In detail, as shown in Figure 4a in the new revision, QPLEX constantly and significantly outperforms baselines over the whole training process and exceeds the best baseline with at least 10% median test win rate averaged across all 17 scenarios. As shown in Figure 4b in the revision, among all 17 tasks, QPLEX is the best performer on up to eight tasks, underperforms on just two tasks, and ties for the best performer with some best baselines on the rest tasks. For example, Figure 5d shows that QPLEX significantly outperforms other baselines in hard map 2c\_vs\_64zg with online data collection. In other simple tasks, QPLEX also significantly outperforms baselines with its high sample efficiency.
>
> [1] *Mahajan A, Rashid T, Samvelyan M, et al. Maven: Multi-agent variational exploration. Advances in Neural Information Processing Systems. 2019: 7613-7624.*
>
> [2] *Samvelyan M, Rashid T, de Witt C S, et al. The starcraft multi-agent challenge. arXiv preprint arXiv:1902.04043, 2019.*

---

> > ### Author Response · Authors · 2020-11-22
> > **Response to Reviewer 2 (Part 2)**
> >
> > **Q5**: What does the strikethrough mean in Fig. 2a (pay off matrix)?
> >
> > Figure 2a shows two matrix games. The strikethrough in Figure 2a shows another matrix game proposed by QTRAN, whose joint reward is zero in the corresponding entries. To investigate the effects of QPLEX’s complete IGM expressiveness on learning optimality, we have designed a harder matrix game illustrated in Figure 2a, which has a harder local optimum, i.e., with the corresponding entries having a joint reward 6. In Appendix C, we also show the learning performance on the matrix game proposed by QTRAN, which can be solved by QPLEX and QTRAN, but not by VDN and QMIX.

---

### Official Review · AnonReviewer4 · 2020-10-28
**Need for further discussion on comparison with QTRAN**

**Rating:** 6
**Confidence:** 4

**Review:**

The authors propose a new value-based method in cooperative multi-agent reinforcement learning. This paper proposes a duplex dueling network that can solve both the limited representation power problem existing in QMIX, and the approximation error problem due to neural network separation, existing in QTRAN. The authors show the novelty of the proposed method through theoretical proof and experimental results. This paper is well written, but there are two significant concerns:

1. The authors argue that QTRAN does not perform well in complex domains because it uses soft constraints, but QPLEX needs to demonstrate this more precisely. QPLEX uses two additional neural network technologies, and It seems the performance gains through these technologies are significant. First, QTRAN uses summation of individual action-values, while QPLEX uses summation of transformed action-values, and QPLEX constructs an advantage function using a scalable multi-head attention module. The authors need to show that performance is better than QTRAN even when these factors are removed. Also, in figure 2(b), QTRAN learns the global optimal policy quickly, but QPLEX learns slowly, so it seems necessary to explain. On the graph, in a matrix game, QTRAN learns more than five times faster than QPLEX.

2. An additional explanation of the authors' environmental selection criteria is needed. In the online data collection setting, the authors use environments such as 5s10z and 7sz, which seem to be environments that were not well used in existing MARL papers, and most environments only use stalker and zealot as agents. Also, the online data collection experiment and the offline data collection experiment use different environments. If the authors use a unified environment, the experiment results will be more enjoyable.

---

> ### Author Response · Authors · 2020-11-22
> **Response to Reviewer 4 (Part 1)**
>
> Thanks for your comments. We provide clarification to your questions and concerns as below. We appreciate if you have any further questions or comments.
>
> **Q1**: QPLEX uses two additional neural network technologies. The authors need to show that performance is better than QTRAN even when these factors are removed.
>
> Without using these two additional neural network techniques, QPLEX still significantly outperforms QTRAN. We conducted an ablation study by removing the *Transformation* module and replacing the multi-head attention module with a simple one-layer forward model in the QPLEX's dueling architecture,  denoted as *QPLEX-wo-trans-atten*. In addition, we also introduce a variant of QTRAN, which also uses the *Transformation* module for individual Q-functions, denoted as *QTRAN-w-trans*.
>
> In Appendix J of this revision, Figure 16 illustrates the averaged median test win rate % over all tested scenarios. Detailed learning curves on these tasks are shown in Figure 17. These empirical results show that QPLEX-wo-trans-atten significantly outperforms QTRAN and QTRAN-w-trans, which implies that the outperformance of QPLEX over QTRAN is largely due to its duplex dueling architecture. It can also be seen that QTRAN-w-trans cannot significantly improve the performance of QTRAN, which implies that QTRAN cannot benefit a lot from an extra *Transformation* module empirically.
>
> We would like to emphasize that, unlike the multi-head attention module, the *Transformation* module is actually a necessary module for QPLEX to realize IGM, as shown in Figure 1 and by the proof of Proposition 2. Therefore, it is actually fair to evaluate QPLEX with the *Transformation* module.
>
> **Q2**: Also, in figure 2(b), QTRAN learns the global optimal policy quickly, but QPLEX learns slowly, so it seems necessary to explain. On the graph, in a matrix game, QTRAN learns more than five times faster than QPLEX.
>
> This is actually an expected result. This is because QTRAN has advantages in simple problems, particularly with 2 or 3 agents. It uses a soft constraint to realize the IGM principle, which can be easily satisfied in the two-agent one-step matrix game. In this case, directly estimating a joint Q-function (which QTRAN does) can achieve fast optimization performance. However, QTRAN generally has a hard time enforcing the IGM constraint in complex problems and thus significantly underperforms in StarCraft II micromanagement benchmark tasks.
>
> In contrast, to enable efficient learning in complex problems, QPLEX encodes the IGM principle into neural networks (a hard constraint of IGM) and uses a duplex dueling architecture that composes individual Q-functions to a joint Q-function without directly estimating the joint Q-function. Although this elaborated value factorization structure needs more gradient steps to optimize the TD loss, it enables QPLEX with the exact realization of IGM even in the complex domains. As discussed in Section 4.3, empirical results on the StarCraft II micromanagement benchmark show that QPLEX significantly outperforms other baselines, while QTRAN performs the worst (see Figure 4) among all baselines in the online data collection setting.

---

> > ### Author Response · Authors · 2020-11-22
> > **Response to Reviewer 4 (Part 2)**
> >
> > **Q3**: An additional explanation of the authors' environmental selection criteria is needed. In the online data collection setting, the authors use environments such as 5s10z and 7sz, which seem to be environments that were not well used in existing MARL papers, and most environments only use stalker and zealot as agents.
> >
> > In addition to the environments shown in the original submission, we have also conducted extensive experiments in more tasks included these new results in Section 4.3 in the new revision. We evaluate QPLEX in 17 scenarios of the StarCraft II benchmark, which contains 14 popular standard maps proposed by SMAC (Samvelyan et al., 2019) and three new super hard cooperative tasks. In this thorough evaluation, QPLEX demonstrates significant outperformance over other baselines, as shown by Figure 4, 5, and 8. In detail, as shown in Figure 4a in the new revision, QPLEX constantly and significantly outperforms baselines over the whole training process and exceeds the best baseline with at least 10% median test win rate averaged across all 17 scenarios. As shown in Figure 4b in the revision, among all 17 tasks, QPLEX is the best performer on up to eight tasks, underperforms on just two tasks, and ties for the best performer with some best baselines on the rest tasks.
> >
> > Zealots and stalkers are popular soldiers when considering challenging coordination in decentralized StarCraft II micromanagement tasks, and SMAC contains many popular maps (2s3z, 3s5z, 3s\_vs\_5z, etc.) using these two Protoss ground units. Zealots are melee soldiers, and stalkers are long-range soldiers. Considering these melee soldiers and long-range soldiers in challenging maps, agents need to learn complex cooperative behaviors to beat the enemies. In Appendix G, we have introduced the visualized strategies of QPLEX and QMIX on the 5s10z map to illustrate the effective coordinated pattern that QPLEX learns.
> >
> > [1] *Samvelyan M, Rashid T, de Witt C S, et al. The starcraft multi-agent challenge. arXiv preprint arXiv:1902.04043, 2019.*

---

> > > ### Comment · AnonReviewer4 · 2020-11-23
> > > **Thank you for your kindly reply. I have additional questions**
> > >
> > > Is there a reason you didn't cite the most recent paper, Weighted QMIX [1], in your paper? Comparison with this paper feels essential. Also in this paper, it says that QPLEX doesn't work well in a predator-prey environment.
> > >
> > > **References**
> > >
> > > [1] Rashid, Tabish, et al. "Weighted QMIX: Expanding Monotonic Value Function Factorisation." NeurIPS 2020

---

> > > > ### Author Response · Authors · 2020-11-24
> > > > **Response to the questions about Weighted QMIX**
> > > >
> > > > Thank you for your feedback.  We provide clarification to your questions and concerns as below. We appreciate if you have any further questions or comments.
> > > >
> > > > **Q1**: Is there a reason you didn't cite the most recent paper, Weighted QMIX [1], in your paper? Comparison with this paper feels essential.
> > > >
> > > > We have conducted an extensive evaluation and compared QPLEX to Weighted QMIX (OW-QMIX and CW-QMIX; Rashid et al., 2020) in the 17 StarCraft II benchmark tasks. Empirical results show that QPLEX  significantly outperforms OW-QMIX and CW-QMIX. Detailed results are shown in Section K.1 of the new revision. As illustrated in Figure 18a in the revision, QPLEX constantly and significantly outperforms Weighted QMIX over the whole training process and exceeds the best baseline with at least 10% median test win rate averaged across all 17 scenarios. As shown in Figure 18b, among all 17 tasks, QPLEX is the best performer on up to eight tasks, underperforms on just two tasks, and ties for the best performer with some best baselines on the rest tasks. Figure 19 shows the individual learning curves on 17 tasks.
> > > >
> > > > Moreover, although OW-QMIX or CW-QMIX outperforms QMIX in some tasks (illustrated in Figure 19), OW-QMIX and CW-QMIX show very similar overall performance as QMIX across 17 StarCraft II benchmark tasks, as illustrated in Figure 18a. In contrast, QPLEX achieves significant improvement in convergence performance in a lot of hard and super hard maps and demonstrates high sample efficiency across most scenarios (see Figure 19).
> > > >
> > > > **Q2**: Also in this paper, it says that QPLEX doesn't work well in a predator-prey environment.
> > > >
> > > > As shown in Figure 20 in Section K.2 in the revision, QPLEX can solve this task by using an $\epsilon$-greedy exploration strategy, which is also discussed in the paper of Weighted QMIX. This strategy's $\epsilon$ is linearly annealed from 1 to 0.05 over 1 million timesteps, increased from the 50k used in SMAC (Samvelyan et al., 2019).
> > > >
> > > > Predator-prey is a multi-agent coordinated game with miscoordination penalties. In order to collect the experience with the positive reward of agents' coordinated actions, extensive exploration can benefit multi-agent Q-learning algorithms to solve this kind of tasks. Weighted QMIX shapes the data distribution with an importance weight to boost efficient learning, which can also be regarded as a type of biased exploration. As shown in Figure 20 in Section K.2, besides Weighted QMIX, QTRAN, and QPLEX, QMIX can also solve this task by using the same $\epsilon$-greedy strategy as QPLEX, but QPLEX enjoys higher sample efficiency than QMIX.
> > > >
> > > > [1] *Rashid T, Farquhar G, Peng B, et al. Weighted QMIX: Expanding Monotonic Value Function Factorisation. arXiv preprint arXiv:2006.10800, 2020.*
> > > >
> > > > [2] *Samvelyan M, Rashid T, de Witt C S, et al. The starcraft multi-agent challenge. arXiv preprint arXiv:1902.04043, 2019.*

---

> > > > > ### Comment · AnonReviewer4 · 2020-11-25
> > > > > **Thanks for your kind responses**
> > > > >
> > > > > I've increased my score. In the future, it would be nice to have a comparison with the Weighted QMIX in the main paper. Including intro, matrix games, and two-step MMDP.

---

> > > > > > ### Author Response · Authors · 2020-11-25
> > > > > > **Thanks for your kind evaluation**
> > > > > >
> > > > > > Thank you very much for your evaluation and additional comments. We will incorporate discussions about Weighted QMIX and extend its experiments to the matrix game and the two-step MMDP in the main paper.

---

### Official Review · AnonReviewer3 · 2020-10-28
**Recommendation to accept**

**Rating:** 6
**Confidence:** 5

**Review:**

##########################################################################

Summary:

The paper proposed a multi-agent Q Learning algorithm with an entire IGM function class for cooperative games. The key idea is to leverage a duplex dueling network architecture to factorize the joint action-value function into individual action-value functions. The main contributions of the work lie in that the proposed method offered an highly scalable algorithms for cooperative tasks. Empirical results show that the method could achieve significant improvement in StarCraftII tasks.
##########################################################################

Reasons for score:

Overall, I'd vote for acceptance to the paper. I like the idea that is using a Transformer to learn to decouple the relationship between different agents. However, the main concerns of mine lie in the details of the proposed model(see cons below).
##########################################################################
Pros:

Overall, the paper is well written. The motivation of the proposed method is straightforward and sound. This paper provides comprehensive experiments, including didactic problems and complex benchmarks, to demonstrate the efficacy and efficiency of the proposed methods.
The scalability of previous methods is one of the most critical issues of a cooperative multi-agent setting. The proposed methods give a practical solution for that.
The credit assignment problem is another exciting problem of a multi-agent system.



##########################################################################
Cons:
 Although the main motivation of the proposed method is to address the scalability of previous methods, some details are unclear to me
Why does \lambda take joint-action and joint-observation as a parameter in eq 10.
What is the computation cost to calculate of \lambda for every agent?
Although the proposed method conducts a few experiments, I still suggest the authors conduct the following ablation studies to enhance the quality of the paper:
Why do the choice of layer and the number of heads are not aligned on matrix and StarCraft II tasks?
The ablation studies on Qatten


##########################################################################

Questions during the rebuttal period:

Please address and clarify the cons above

---

> ### Author Response · Authors · 2020-11-22
> **Response to Reviewer 3**
>
> Thanks for your inspiring comments. We provide clarification to your questions as below. We appreciate if you have any further questions or comments.
>
> **Q1**: Why does $\lambda$ take joint-action and joint-observation as a parameter in eq 10.
>
> In theory, making importance weight $\lambda$ depend on joint-action and joint-observation is important for QPLEX to realize a complete IGM function class (see Proposition 2). In practice, the positivity constraint of $\lambda$ can be easily realized by a neural network, whose computation is efficient (please refer to Q2 for details).
>
> In detail, to achieve the necessary condition of the IGM principle, the joint action-value function needs to take joint-action and joint-observation into account. However, directly approximate the joint Q-function (which QTRAN does) cannot efficiently support scalable argmax operator on the exponentially large joint-action space (which multi-agent Q-learning needs).
>
> By introducing duplex dueling architecture, QPLEX can realize a scalable and accurate calculation of the argmax operator using individual action-value functions and support joint Q-function with joint-action and joint-observation by feeding this joint information into importance weights $\lambda$. Note that such expressiveness of $\lambda$ will not only keep the consistency between joint and local action selections (see Fact 1 and Proposition 2) but also provide QPLEX's joint Q-function with joint-action and joint-observation to achieve a complete IGM function class theoretically. From an empirical perspective, the neural network implementation of $\lambda$ with joint-action and joint-observation can be regarded as to effectively approximate the one-step TD target (which also contains the same amount of entries with joint-action and joint-observation) for better sample efficiency.
>
> **Q2**: What is the computation cost to calculate of $\lambda$ for every agent?
>
> Calculating $\lambda$ for all agents is scalable and efficient. QPLEX uses a multi-head attention structure to implement importance weights $\lambda$, which means that we can use one forward pass of neural networks to calculate $\lambda$ for all agents during the end-to-end centralized training. During the decentralized execution or the greedy joint action selection, as mentioned in the *Individual Action-Value Function* component in Section 3.2, each agent can make their own decisions on individual action-value functions (due to the consistency between greedy local and joint selection) and does not need to calculate $\lambda$ in these cases.
>
> **Q3**: Why do the choice of layer and the number of heads are not aligned on matrix and StarCraft II tasks?
>
> We trade off the expressiveness and learning efficiency of the multi-head attention network module for estimating $\lambda$. For StarCraft II tasks, QPLEX used a multi-head attention just with one layer and four heads. We conducted an ablation study by increasing the number of heads and layers, as discussed in Appendix I in the new revision. As shown in Figure 14, using more heads does not change the performance, but using more layers may slightly degenerate the learning efficiency of QPLEX in this complex domain. This is because using more layers significantly increase the number of parameters and requires more samples for learning, which may offset the benefits of higher expressiveness it brings. This ablation study shows that a simple attention neural network with just one layer and four heads has enough expressiveness to handle these StarCraft II tasks.
>
> However, in some didactic corner-cases with an adequate and uniform dataset, a harder matrix game illustrated in Figure 2a requires very high precision in estimating the action-value function in order to differentiate the optimal solution from the sub-optimal solutions. For this matrix game, a multi-head attention structure with more layers and heads has a more representational capacity and results in better performance, as demonstrated by the ablation study illustrated in Figure 2c in Section 4.1.
>
> In general, it is sufficient and efficient for QPLEX to use a simple multi-head attention with a small number of heads and layers (as evaluated in StarCraft II tasks).
>
> **Q4**: Additional experimental results.
>
> For your reference, we have also conducted extensive experiments in more tasks and additional ablation studies and included these results in Section 4.3 in the revision. We evaluate QPLEX in 17 scenarios of the StarCraft II benchmark, which contains 14 popular standard maps proposed by SMAC (Samvelyan et al., 2019) and three new super hard cooperative tasks. These new results further demonstrate the advantages of QPLEX over state-of-the-art baselines.
>
> [1] *Samvelyan M, Rashid T, de Witt C S, et al. The starcraft multi-agent challenge. arXiv preprint arXiv:1902.04043, 2019.*

---

### Official Review · AnonReviewer1 · 2020-10-29
**Interesting paper.**

**Rating:** 7
**Confidence:** 2

**Review:**

This paper presents an approach towards multi-agent reinforcement learning (MARL) in the context of centralized training with decentralized execution (CTDE). This paper considers the Individual Global Max (IGM) principle that involves consistency between the local action of every agent and joint action of all agents.

The paper presents an approach that enforces IGM through an appropriate selection of a neural network architecture that transforms the IGM principle into constraints on the output range of an appropriate neural network. These constraints enable enforcing consistency that is necessary for enforcing the IGM principle.

Overall, I find the paper to be well written, and is well motivated. I went through the theoretical claims but haven’t been able to completely verify the claims of Proposition 2. In terms of experimental results, the paper presents results on a variant of a matrix game considered by prior work, two state MMDP and in both online and offline settings of a micro-management benchmark of StarCraft.

While I believe the paper presents good contributions to MARL, I am not an expert in this area, so, I will mark my review with low confidence since I am unable to offer constructive pointers on possible improvements etc.

---

> ### Author Response · Authors · 2020-11-22
> **Response to Reviewer 1**
>
> Thanks for your thoughtful comments.
>
> For your reference, we have also conducted extensive experiments in more tasks and additional ablation studies and included these results in Section 4.3 in the revision.  These new results further demonstrate the advantages of QPLEX over state-of-the-art baselines.
>
> We evaluate QPLEX in 17 scenarios of the StarCraft II benchmark, which contains 14 popular standard maps proposed by SMAC (Samvelyan et al., 2019) and three new super hard cooperative tasks. As shown in Figure 4a in the revision, compared with other baselines, QPLEX constantly and significantly outperforms baselines over the whole training process and exceeds the best baseline with at least 10% median test win rate averaged across all 17 scenarios. As shown in Figure 4b in the revision, among all 17 tasks, QPLEX is the best performer on up to eight tasks, underperforms on just two tasks, and ties for the best performer with some best baselines on the rest tasks.
>
> Moreover, we also conduct extra ablation studies, as discussed in Section 4.3.2 and Appendix E, F, I, and J. These ablation results demonstrate the effectiveness of the duplex dueling architecture of QPLEX.
>
> [1] *Samvelyan M, Rashid T, de Witt C S, et al. The starcraft multi-agent challenge. arXiv preprint arXiv:1902.04043, 2019.*

---

> > ### Comment · AnonReviewer1 · 2020-11-23
> > **Reponse to author feedback**
> >
> > Thank you for your response.
> >
> > Based on the author feedback, I will retain the score I have presented above.

---

### Author Response · Authors · 2020-11-23
**Summary of the Revision**

Thank you to all the reviewers for your thoughtful and inspiring comments, which helped us improve our work. Here is a summary of major updates made to the revision:

1. Conducted more extensive experiments on 17 scenarios of the StarCraft II benchmark, which contains 14 popular standard maps proposed by SMAC (Samvelyan et al., 2019) and three new super hard cooperative tasks. These experiments strengthen the outperformance result of QPLEX over state-of-the-art baselines, shown and discussed in Section 4.3.1 and Appendix D.

2. Added new plots showing the overall performance of QPLEX and baselines, averaged across 17 StarCraft II benchmark tasks, illustrated in Figure 4.

3. Added four extra ablation studies to demonstrate further the effectiveness of the duplex dueling architecture of QPLEX, as discussed in Section 4.3.2 and Appendix E, F, I, and J.

We hope that our response and revision address your concerns and questions. We are happy to provide further clarification if you have any additional concerns or comments.

[1] *Samvelyan M, Rashid T, de Witt C S, et al. The starcraft multi-agent challenge. arXiv preprint arXiv:1902.04043, 2019.*

---

### Decision · Program_Chairs · 2021-01-07
**Final Decision**

**Decision:**

Accept (Poster)

**Comment:**

The majority of reviewers recommend acceptance for this paper, and the average score is in the acceptance rate. Only one reviewer (reviewer 2) recommend rejection, and from the reading the review, the authors answer, and the paper, I think it is possible that the reviewer missed the motivation behind this architecture, which is partly reason for rejection. Unfortunately the reviewer did not answer the authors so I cannot be sure if the reviewer is aware of that. Therefore, I am confident in my recommendation to follow the majority of the reviewers.

The reviewers generally believe this paper is well written. The paper has a good structure and although quite technical, is still easy to follow.

Concerns were raised about the scale of the experiments and motivations for some experimental choices.
I appreciate that the authors have extended the set of evaluations on the Starcraft task, and added ablations studies, which partly address some of these concerns.